



# Pore network modeling as a new tool for determining gas diffusivity in peat

Petri Kiuru[1], Marjo Palviainen[2], Arianna Marchionne[3], Tiia Grönholm[4], Maarit Raivonen[5], Lukas Kohl[6,7], and Annamari Laurén[1]

[1]School of Forest Sciences, Faculty of Science and Forestry, University of Eastern Finland, P.O. Box 111, 80101 Joensuu, Finland
[2]Department of Forest Sciences, University of Helsinki, P.O. Box 27, 00014 Helsinki, Finland
[3]Department of Mathematics and Statistics, University of Helsinki, P.O. Box 68, 00014 Helsinki, Finland
[4]Finnish Meteorological Institute (FMI), Erik Palménin aukio 1, 00560 Helsinki, Finland
[5]Institute for Atmospheric and Earth System Research (INAR)/Physics, Faculty of Science, University of Helsinki, P.O. Box 68, 00014 Helsinki, Finland
[6]Department of Agricultural Sciences, University of Helsinki, P.O. Box 56, 00014 Helsinki, Finland
[7]Institute for Atmospheric and Earth System Research (INAR)/Forest Sciences, Faculty of Agriculture and Forestry, University of Helsinki, P.O. Box 56, 00014 Helsinki, Finland

**Correspondence:** Petri Kiuru (petri.kiuru@uef.fi)

**Abstract.** Peatlands are globally significant carbon stocks and may become major sources of greenhouse gasses (GHG) carbon dioxide and methane in changing climate and under anthropogenic management pressure. Diffusion is the dominant gas transport mechanism in peat, and therefore, a proper knowledge of the soil gas diffusion coefficient is important for the estimation of GHG emissions from peatlands. Pore network modeling (PNM) is a potential tool for the determination of gas diffusivity

5   in peat, as it explicitly connects the peat microstructure and the characteristics of the peat pore network to macroscopic gas transport properties. In the present work, we extracted macropore networks from three-dimensional X-ray micro-computed tomography (µCT) images of peat samples and simulated gas diffusion along the networks using PNM. These results were compared to soil gas diffusion coefficients determined from the same samples in the laboratory using the diffusion chamber method. The measurements and simulations were conducted for peat samples from three depths. The soil gas diffusion coeffi-

10  cients were determined under varying water contents adjusted in a pressure plate apparatus. We also assessed the applicability of commonly used gas diffusivity models to peat. The laboratory measurements showed a decrease in gas diffusivity with depth due to a decrease in air-filled porosity and pore space connectivity. However, gas diffusivity remained relatively high close to saturation, which may indicate that the structure of the macropore network is such that it enables the presence of connected diffusion pathways through the peat matrix even in wet conditions. The gas diffusivity models were not very successful in

15  predicting the soil gas diffusion coefficient. This may indicate that the microstructure of peat differs considerably from the structure of mineral soils and other kinds of porous materials, for which the models have been constructed and calibrated. By contrast, the pore network simulations reproduced the laboratory-determined soil gas diffusion coefficients rather well. Thus, the combination of the µCT and PNM methods may offer a promising alternative to the traditional estimation of soil gas diffusivity through laboratory measurements.





## 1 Introduction

Peatlands have an important role in global carbon (C) cycling. Approximately 600 Gt of C is stored in peatlands as peat (Yu et al., 2010), which accounts for one-fifth of the total pool of soil organic C (Leifeld and Menichetti, 2018) and corresponds to more than a half of the C currently held in the atmosphere as carbon dioxide ($CO_2$) (Limpens et al., 2008). Peatlands are vulnerable to management practices and changes in climate and may therefore become one of the major sources of greenhouse gasses (GHGs) $CO_2$ and methane ($CH_4$) in the global C cycle (Frolking et al., 2011; Leifeld et al., 2019). Drainage and lowering the water table (WT) increase the net $CO_2$ emissions of peatlands and may turn them from sinks into sources of $CO_2$ (Ojanen and Minkkinen, 2019; Günther et al., 2020). Many boreal peatlands may also become $CO_2$ sources due to rapid climate warming (Qiu et al., 2020). Despite its low atmospheric concentration, $CH_4$ accounts for about one-fifth of the global radiative forcing and is therefore the second most important anthropogenic GHG after $CO_2$ (Saunois et al., 2020). Peatlands are significant natural sources of $CH_4$ (Lai, 2009; Abdalla et al., 2016; Tsuruta et al., 2019) since about 30 % of global $CH_4$ emissions originate from peatlands or other wetlands (Kirschke et al., 2013).

The emissions of $CO_2$ and $CH_4$ from peatlands are both tightly linked to the gas transport properties of peat, which control the transport of oxygen ($O_2$) into peat profiles and the transport of $CO_2$ and $CH_4$ from peat to the atmosphere. The availability of $O_2$, in turn, controls the decomposition pathway of organic matter in peat (Estop-Aragonés et al., 2012). $O_2$ is transported from the atmosphere into the peat, and it is continuously consumed by heterotrophic respiration, which produces $CO_2$. When the rate of $O_2$ consumption is higher than its supply, the microbial metabolism changes to other electron acceptors, and finally the production of $CH_4$ starts (Bridgham et al., 2013). This is the case below the WT, where anaerobic conditions dominate permanently (Abdalla et al., 2016), but anaerobic conditions may also occur in unsaturated peat if the peat structure does not favor $O_2$ transport (Fan et al., 2014).

Soil pore architecture is a fundamental physical characteristic of soil, and it controls many important soil functions such as water and gas transport and biogeochemical processes (Rabot et al., 2018; Schlüter et al., 2020). Diffusion is considered to be the primary gas transport mechanism in soils (Jin and Jury, 1996; Maier et al., 2020). Because the diffusion coefficients of $O_2$, $CO_2$, and $CH_4$ in air are 4 orders of magnitude higher than in water (Ball and Smith, 2001), the gas diffusion capability of unsaturated soil is closely connected to the size and number of air-filled pores and the connections and pathways between the air-filled pores and the atmosphere. Because the distance of the WT from the soil surface is generally less than 1 m in peatlands (Sarkkola et al., 2010), small pores remain permanently filled with water. Macropores, defined as pores with an effective diameter of greater than 100 μm (Beven and Germann, 1982), are dominant conduits for gas transport and important in many other soil functions (Reddy and DeLaune, 2008; McCarter et al., 2020). Peat macroporosity is generally highest near the surface because the degree of decomposition typically increases with depth and because the decomposition results in decreasing pore volumes (Päivänen, 1973). Macropores form a complex network with open and connected, dead-ended, and isolated individual pores (Rezanezhad et al., 2016). The connectivity of the macropore network regulates its transport properties (Koestel et al., 2020). Pores that are disconnected from the active, interconnected network do not contribute to transport processes. Soil gas diffusivity decreases with increasing soil water content, as pores are removed from the transport



network when they become water-filled (Moldrup et al., 2004). Hysteresis, that is, the difference in soil water content at a
specific water potential between drying and wetting, also affects the size and connectivity of the active pore network (Kiuru
et al., 2022a).

The soil gas diffusion coefficient $D_s$ depends on the air-filled porosity of the soil and the structural properties of the soil pore
network (Jin and Jury, 1996). The relative gas diffusion coefficient or relative diffusivity ($D_s/D_0$), where $D_0$ is the gas diffusion
coefficient in free air, is a gas-independent way to express the soil gas diffusion capability. A variety of models for $D_s/D_0$ have
been developed and used to explore the relationship between the gas diffusivity and the physical properties of porous solids in
general and of mineral soils in particular (e.g., Penman, 1940; Millington, 1959; Millington and Quirk, 1961; Campbell, 1985;
Moldrup et al., 2004). Gas diffusivity models are also needed in process-based models describing biogeochemical processes
and GHG production and emission in soils (Blagodatsky and Smith, 2012; Xu et al., 2016). However, the applicability of
the models to peat is questionable because of the high porosity and other unique physical characteristics of peat (Iiyama and
Hasegawa, 2005). To date, the gas diffusivity of peat has only been investigated in few studies (King and Smith, 1987; Boon
et al., 2013; Hamamoto et al., 2016a), and the rates of gas transport processes in peatlands remain poorly understood.

Soil structure characteristics are generally taken account of in gas diffusivity models by incorporating semiempirical tortuosity–
connectivity factors (Hamamoto et al., 2016b). X-ray micro-computed tomography (μCT) is a noninvasive and nondestructive
imaging technique that can be used for explicit three-dimensional visualization and description of soil structure and pore ar-
chitecture (Helliwell et al., 2013). Extracting a pore network from the μCT images allows the determination of the size of
individual pores and their connectivity (e.g., Dong and Blunt, 2009; Gostick, 2017). Such data enable the use of pore net-
work modeling (PNM), which can be used to simulate gas diffusion through soil and determine $D_s$ (Steele and Nieber, 1994;
Gharedaghloo et al., 2018). Instead of simulating transport processes with direct numerical methods using the image void struc-
tures as the computational mesh, PNM simulates transport in a network of simplified pores, which are constructed on the basis
of the actual pore geometry (Zhao et al., 2020). A great advantage of PNM is that it requires substantially less computational
capacity than the direct simulation methods (Blunt et al., 2013; Xiong et al., 2016). Thus, the combination of μCT and PNM
is a useful method for estimating soil gas diffusivity from the soil structural characteristics. However, few attempts have been
made to apply this promising approach to peat (Gharedaghloo et al., 2018).

In this study, we determined the soil gas diffusion coefficients of peat samples that were collected from a boreal forested
peatland. The objectives of this work were (1) to study the variation in the relative diffusivity of peat with depth and between
different soil water content conditions; (2) to assess the capability of PNM to estimate and characterize the gas diffusion
dynamics in peat; and finally (3) to investigate the applicability of widely used models for gas diffusivity to peat.

## 2   Materials and methods

### 2.1   Field sampling

Peat samples were collected from a forested peatland site in southern Finland (60° 38' N, 23° 57' E, Lettosuo, Tammela). The
site was drained in 1969 with parallel ditch drains in 40 m spacing. The long-term mean annual temperature and precipitation





are 5.2 °C and 621 mm, respectively (Jokinen et al., 2021). The soil type is Histosol and the peat type is *Carex* peat. The site was originally a mesotrophic fen classified as an herb-rich tall-sedge birch–pine fen (Laine and Vasander, 1996). The forest stand is dominated by Scots pine (*Pinus sylvestris* L.) and downy birch (*Betula pubescens* Ehrh.). The dominant height of the trees is 20 m. The understory is composed of Norway spruce (*Picea abies* Karst.). The total stand volume is 230 $m^3$ $ha^{-1}$ and stocking is 2200 stems $ha^{-1}$. The ground vegetation is composed of dwarf shrubs with a coverage of 4 % (*Vaccinium myrtillus* L., *V. vitis-idaea* L.) and herbs (coverage 10.6 %). A detailed site description is available in Kiuru et al. (2022a).

Undisturbed peat samples were collected from seven randomly located pits and three different depths (0–5 cm, 20–25 cm, and 40–45 cm, later referred to as the top, middle, and bottom layer, respectively). A pit with a depth of 50–60 cm with an undisturbed vertical face was dug, after which the profile depth was measured with a ruler. Vertically oriented peat samples were extracted along the pit face into acrylic cylinders (diameter 50 cm, height 50 cm) using a sharp knife and scissors, paying attention to keeping the peat structure undisturbed.

### 2.2 Gas diffusivity measurement

Firstly, the undisturbed samples were saturated for 1 d at a constant temperature of 20 °C. Next, the samples were placed inside a pressure chamber in contact with a ceramic porous plate and dehydrated by applying the pressures of 1, 3, 6, and 10 kPa. Following equilibrium between the soil matric potential and the applied pressure, the soil samples were removed from the pressure chamber and weighed. The gas diffusivity measurement was then performed for each sample. The procedure was repeated for each moisture equilibrium. The sample volumes decreased upon drying during the experiment. At the end of the experiment, the sample height and diameter were measured to determine the shrinkage. The water retention measurement procedure is described in more detail in Kiuru et al. (2022a).

The bulk density of a peat sample was determined from its dry mass and the saturated volume. The volumetric water content at each matric potential was also calculated in relation to the saturated volume. Total porosity was estimated from the bulk density and a mean particle density for organic soil of 1500 $kg\,m^{-3}$ (Redding and Devito, 2006). Air-filled porosity was determined at each state as the difference between total porosity and the respective volumetric water content.

In order to determine the soil gas diffusivity values $D_s$, each peat sample was attached to a gas diffusion chamber (Currie, 1960; Edling, 1986). The diffusion chambers consisted of an empty cylindrical head space (diameter 50 mm, length 100 mm) composed of acrylic material that was air-tightly connected to the peat core. Each diffusion chamber was equipped with two silicone tubes through which nitrogen gas ($N_2$) was circulated through the head space until the $N_2$ content was around 99 %, after which the tubes were closed. This set a concentration difference between the upper and lower ends of the peat core, and $N_2$ started to diffuse out from the head space through the peat. $N_2$ was chosen because it is nontoxic and was neither produced nor consumed in significant quantities during the 3 hour measurement. We measured the changes in $N_2$ concentration inside the diffusion chamber with a gas syringe inserted through a rubber membrane located at the bottom of the chamber.

Two gas samples were extracted from the head space through a rubber septum using a needle and a syringe during the experiment approximately 45 and 120 min after sealing the chamber. The gas samples (8 mL) were injected into helium-flushed pre-evacuated 3 mL Exetainer vials (Labco, UK), and $N_2$, $O_2$, and $CO_2$ concentrations were quantified using a gas





chromatograph (Agilent 7890B) equipped with a thermal conductivity detector (Soinne et al., 2021). Samples were injected using a 0.5 mL loop and separated on a HayeSep Q 80/100 column (Agilent G3591-81004; 3 ft by $\frac{1}{8}$ in) followed by a HP-PLOT Molesieve column (Agilent 19095P-MS0; 50 m by 0.53 mm by 50 µm). Both columns were held isothermally at 40 °C. Helium was used as a carrier gas at constant pressure (30 psi) resulting in a flow rate of 16.3 mL min$^{-1}$. The system was

calibrated with a gas mixture that contained 5 % $CO_2$, 16 % $O_2$, and 79 % $N_2$.

The soil gas diffusion coefficient ($D_s$, m$^2$ s$^{-1}$) was calculated using the approximate formula (Bakker and Hidding, 1970):

$$D_s = \frac{l_c l_s}{(t_2 - t_1)} \ln\left(\frac{\Delta C_1}{\Delta C_2}\right)\left(1 + 0.34 a \frac{l_s}{l_c}\right) \tag{1}$$

where $a$ is the air-filled porosity (m$^3$ m$^{-3}$) of the peat sample, $l_s$ (m) is the length of the sample, $l_c$ (m) is the length of the diffusion chamber head space, and $\Delta C_1$ and $\Delta C_2$ (mol m$^{-3}$) are the gas concentration differences between the top and bottom

of the sample at times $t_1$ and $t_2$ (s), respectively. The mole fraction of $N_2$ in free air was assumed to be 78 %. Of a total of 84 diffusion measurements, 16 were discarded because of inconsistent $N_2$ concentration values that were most probably caused by leakages in the diffusion measurement system or during gas sampling.

## 2.3 Comparison with existing models

We compared the calculated relative diffusivity ($D_s/D_0$) values with values obtained from commonly used gas diffusivity

models by Currie (1960) and Millington and Quirk (1961), another model by Millington and Quirk (1960), and a more complex model by Moldrup et al. (2004). The models relate the relative diffusivity to air-filled porosity $a$ and total porosity $\varepsilon$ (m$^3$ m$^{-3}$). The model by Currie (1960) (hereafter referred to as the CC model) uses only the value of air-filled porosity to determine the relative diffusivity as

$$\frac{D_s}{D_0} = \alpha a^\beta \tag{2}$$

The parameters $\alpha$ and $\beta$ can be regarded as relating to pore tortuosity and the pore size distribution, respectively. Commonly used values for soil with a high water content, $\alpha = 0.9$ and $\beta = 2.3$, were suggested by Campbell (1985). The model by Millington and Quirk (1961) (MQ61) is a simple nonlinear model that also requires the value of total porosity to predict the $D_s(a)$ curve of soil:

$$\frac{D_s}{D_0} = \frac{a^{10/3}}{\varepsilon^2} \tag{3}$$

Millington and Quirk (1960) also proposed the model (MQ60)

$$\frac{D_s}{D_0} = \frac{a^2}{\varepsilon^{2/3}} \tag{4}$$

which has been shown to outperform the MQ61 model in some cases (Washington et al., 1994; Jin and Jury, 1996). The three-porosity model (TPM) presented by Moldrup et al. (2004) is given by

$$\frac{D_s}{D_0} = \varepsilon^2 \left(\frac{a}{\varepsilon}\right)^X \tag{5}$$



In addition to $a$ and $\varepsilon$, it requires a third porosity value $a_{100}$, air-filled porosity at $-100\ \mathrm{cmH_2O}$ ($-10\ \mathrm{kPa}$) matric potential. The parameter $X$ describes the tortuosity and connectivity of soil, and it is obtained by assuming an empirically developed relationship between the soil gas diffusion coefficient at $-100\ \mathrm{cmH_2O}$ matric potential ($D_{100}$) and the corresponding air-filled porosity (Moldrup et al., 2000):

$$\frac{D_{100}}{D_0} = 2a_{100}^3 + 0.04a_{100} \tag{6}$$

The parameter X is obtained by combining Eqs. (5) and (6) as

$$X = \frac{\log\left[\left(2a_{100}^3 + 0.04a_{100}\right)/\varepsilon^2\right]}{\log\left(a_{100}/\varepsilon\right)} \tag{7}$$

The TPM model has been shown to give accurate predictions of $D_s(a)$ curves across soil types and total porosities (Moldrup et al., 2004). The diffusion coefficient of $N_2$ in free air was assumed to equal the diffusion coefficient of $O_2$ in $N_2$ at 20 °C, $0.202\ \mathrm{cm^2\,s^{-1}}$ (Rumble, 2021).

## 2.4 Pore network imaging and analyses

In addition to laboratory experiments, we studied how the pore network characteristics affect gas diffusion in peat using pore network simulations. The pore networks were extracted from μCT images taken from the same peat samples as used in the laboratory experiment, and the networks were used as the gas transfer domains in the simulations.

### 2.4.1 μCT imaging and image processing

The peat samples, which were at $-10\ \mathrm{kPa}$ matric potential, were scanned in the micro-CT laboratory in the University of Helsinki with the GE Phoenix Nanotom system after the water retention experiment. The final voxel (cubic 3D image element) size after image reconstruction was 50 μm, and the resulting 3D images were 1142 by 1142 by 1152 voxels in size. The image preprocessing stage, including straightening, cropping, noise filtering, and binary segmentation, was performed using the Python image processing packages SciPy ndimage (Virtanen et al., 2020) and scikit-image (van der Walt et al., 2014) and the image analysis toolkit PoreSpy (Gostick et al., 2019). The images were segmented into void (air) and solid (water and organic material) volumes using the widely utilized global thresholding method by Otsu (1979). The final size of each binarized image was 1000 by 1000 by 1000 voxels, of which a cylindrical peat volume with a height of 1000 voxels and a diameter of 1000 voxels was selected for further analysis.

### 2.4.2 Pore networks

The extraction of pore networks from the $1000^3$-voxel binary images and the determination of the pore network geometry were performed using a marker-based watershed segmentation method (Gostick, 2017) available in PoreSpy. Because the feature resolution of μCT-generated images is generally about twice the voxel size (Stock, 2008), the size of the smallest distinguishable feature was 100 μm. The extracted pore system can be divided into clusters of interconnected pores and a group of isolated





pores. The largest of these clusters, which was assumed to be the only cluster that extends through the applied sample domain,

was defined as the pore network. Network porosity was defined as the ratio of the combined volume of the pores in the network to the total volume of the network domain.

Pore volume was determined by the number of voxels in the pore region, and pore diameter was defined as the diameter of the largest sphere that fits inside the pore region. Similarly, throat diameter was defined as the diameter of the largest circle that fits inside the throat region. A stick-and-ball simplification of the pore network geometry was employed: the pores

were considered as spheres centered at the centroids of the pore regions, and the throats were cylindrical tubes connecting adjacent pores. Because the centroids of two interconnected pores and the centroid of the throat between them were generally not collinear, the conduit length $d$ between neighboring pores was determined as the sum of the distances between each pore centroid ($p_a$ and $p_b$) and the throat centroid (t): $d(p_a, p_b) = d(p_a, t) + d(t, p_b)$. A more detailed description of the workflow from the µCT imaging to pore network extraction is found in Kiuru et al. (2022a).

**2.5  Diffusion simulation with PNM**

We used the open-source pore network modeling package OpenPNM (Gostick et al., 2016) for the simulation of water retention and gas diffusion through a pore network. OpenPNM has been recently used for the simulation of fluid flow in rocks, sediments, and mineral soil (e.g., Merey, 2019; Yang et al., 2019; Dong et al., 2021) but not for organic porous matter. The water retention simulation determined the extent and configuration of the air-filled pore network at each external pressure step, and diffusion

simulation was then used to determine the effective diffusivity of the pore network. The maximum external pressure applied in the simulations, 2.88 kPa, was determined by the minimum throat diameter (100 µm). The water retention simulation was performed employing the algorithm for drainage percolation in OpenPNM (see Kiuru et al., 2022a). The diffusion simulation gives the rate of the steady-state diffusion of a gas through an air-filled pore network. Mass conservation is required in each pore in the network, and the mass transfer between each pair of pores is determined by the Fick's first law of diffusion. The

effective diffusivity of the pore network, which can be interpreted as the simulated soil gas diffusion coefficient $D_{pnm}$, can finally be obtained using the Fick's first law of diffusion as

$$D_{pnm} = \frac{NL}{A \Delta C} \tag{8}$$

where $N$ (mol s$^{-1}$) is the rate of steady-state diffusion through the network, $L$ (m) is the length of the network domain, $A$ (m$^2$) is the cross-sectional area of the domain, and $\Delta C$ (mol m$^{-3}$) is the concentration gradient between the opposite boundaries of

the domain in the direction of the flow.

The network domain size used in the simulations had to be as close to the total sample size as possible so that comparison with the measured soil gas diffusion coefficients would be reasonable. Thus, no more than 100 voxels, corresponding to 5 mm slices, at the top and bottom of the sample images were ignored so that the influence of the roughness of the sample surfaces and the image reconstruction defects near the image boundaries were still excluded. The network domain was therefore 40 mm

in height, and it included the whole cylindrical region in the horizontal direction. Because some of the top-layer samples had shrunk slightly in the vertical direction during the water retention experiment, the network domain height was reduced to 30





or 35 mm as needed. The resulting image was thereafter divided into four similar-shaped regions with horizontal dimensions of 500 by 500 voxels. A separate pore network was extracted for each region with PoreSpy. Diffusion simulations for the four subnetworks were performed using the same pressure steps in each simulation. The total effective diffusivity at each pressure

step for the cylindrical network domain with a height of 800 voxels and a diameter of 1000 voxels was then calculated from the combined diffusion rates of the subnetworks and the cross-sectional area of the total network domain.

## 2.6 Statistics

We applied a one-way analysis of variance (ANOVA) followed by Tukey's pairwise multiple comparison test to determine the possible influence of depth on the soil gas diffusion coefficient. If residual normality and variance homogeneity could not be

assumed, an independent sample $t$ test was applied instead. A paired sample $t$ test was applied to analyze the difference between the measured and simulated soil gas diffusion coefficients. The statistical analyses were conducted with the statistical function module in SciPy and the Python packages statsmodels (Seabold and Perktold, 2010) and hypothetical (Schlegel, 2020).

We also assessed the performance of the pore network simulations against the measured soil gas diffusion coefficients using a Bland–Altman plot (Bland and Altman, 1999; Giavarina, 2015). It is a graphical technique for assessing the agreement between

values ($X$ and $Y$) obtained from two different measurement methods. The Bland–Altman plot is constructed by plotting the differences between each pair of values ($X_i - Y_i$) against the averages of the pairs of values ($(X_i + Y_i)/2$). An estimated agreement interval, which is bounded by the limits of agreement and covers 95 % of the difference range, is also shown in the plot.

We examined the agreement between measured and model-estimated relative diffusivity with several performance metrics.

The coefficient of determination is defined as

$$R^2 = 1 - \frac{SS_{\mathrm{res}}}{SS_{\mathrm{tot}}} \tag{9}$$

where $SS_{\mathrm{res}}$ is the sum of the squares of the residuals between measured values $D_{\mathrm{mea}}$ and model-estimated values $D_{\mathrm{mod}}$:

$$SS_{\mathrm{res}} = \sum_i \left(D_{\mathrm{mod},\,i} - D_{\mathrm{mea},\,i}\right)^2 \tag{10}$$

and $SS_{\mathrm{tot}}$ is the sum over the squared differences between the measurements and their mean $\overline{D_{\mathrm{mea}}}$:

$$SS_{\mathrm{tot}} = \sum_i \left(D_{\mathrm{mea},\,i} - \overline{D_{\mathrm{mea}}}\right)^2 \tag{11}$$

Lin's concordance correlation coefficient $\rho_{\mathrm{c}}$ (Lin, 1989) assesses the degree of agreement between two continuous variables taking into account both the linear relationship between the two variables and the deviation from the perfect agreement line (Dhanoa et al., 1999). The value of $\rho_{\mathrm{c}}$ ranges from 0 (no concordance) to 1 (perfect agreement). It is calculated as

$$\rho_{\mathrm{c}} = \frac{2\sigma_{\mathrm{ms}}}{\sigma_{\mathrm{m}}^2 + \sigma_{\mathrm{s}}^2 + (\mu_{\mathrm{m}} - \mu_{\mathrm{s}})^2} \tag{12}$$

where $\mu_{\mathrm{m}}$ and $\mu_{\mathrm{s}}$ are the means of measured and model-estimated values, respectively, $\sigma_{\mathrm{m}}$ and $\sigma_{\mathrm{s}}$ are the corresponding variances, and $\sigma_{\mathrm{ms}}$ is the covariance between measured and model-estimated values.





The Akaike information criterion (AIC) (Akaike, 1974) takes into account the number of model parameters in performance comparison. AIC with a correction for small sample size is defined as (Burnham and Anderson, 2004)

$$\mathrm{AIC_c} = n\ln\left(\frac{SS_{\mathrm{res}}}{n}\right) + 2K + \frac{2K\left(K+1\right)}{n-K-1} \tag{13}$$

where $n$ is the sample size and $K$ is the number of model parameters. AIC can only be used for a relative ranking of different models according to their performance (Tozzi et al., 2020). A smaller or more negative AIC indicates better model performance. The relative performance can be characterized by rescaling to

$$\Delta\mathrm{AIC}_i = \mathrm{AIC}_i - \mathrm{AIC}_{\mathrm{min}} \tag{14}$$

where $\mathrm{AIC}_{\mathrm{min}}$ is the minimum of the AIC values for $i$ different models.

## 250  3  Results

### 3.1  Soil gas diffusivity and air-filled porosity measurements

The air-filled porosities of the samples with successful gas diffusivity measurements differed significantly between the layers (ANOVA, $p < 0.05$, Table 1). Air-filled porosity was highest in the top layer and decreased with depth. The difference between the middle and bottom layers was, however, nonsignificant at all matric potentials (Tukey's post hoc test, $p = 0.15 - 0.22$) other

than $-1\,\mathrm{kPa}$ (independent sample $t$ test, $t(11) = 4.15$ and $p = 0.002$). Air-filled porosity varied the most between samples in the bottom layer (relative standard deviation (RSD) at different matric potentials between 0.27 and 0.62) followed by the middle (RSD 0.24–0.37) and the top layer (RSD 0.08–0.44).

The soil gas diffusion coefficients of the peat samples also differed significantly between the sampling depths (ANOVA, $p < 0.05$, Table 2). Similarly to air-filled porosity, the soil gas diffusion coefficient was highest in the top layer at all matric

potentials and decreased with depth. However, no significant difference was observed between the middle and bottom layers (Tukey's post hoc test, $p = 0.9$). ANOVA was not applicable to the $-1\,\mathrm{kPa}$ samples, but there was no significant difference in $D_{\mathrm{s}}$ between the middle and bottom layers (independent sample $t$ test, $t(11) = 1.19$ and $p = 0.26$). The variation in $D_{\mathrm{s}}$ between samples was largest in the bottom (RSD 0.34–0.66) and top (RSD 0.20–0.58) layers but also considerable in the middle layer (RSD 0.16–0.40).

While both air-filled porosity and the soil gas diffusion coefficient decreased with depth, they exhibited distinct depth profiles. The decrease in gas diffusivity between the top layer and the middle layer was generally slightly greater than the corresponding decrease in air-filled porosity. This was most pronounced at $-6\,\mathrm{kPa}$ conditions: the soil gas diffusion coefficient decreased by 65 %, whereas the decrease in air-filled porosity was less than 50 %. By contrast, the difference in the average soil gas diffusion coefficient between the middle and bottom layers was smaller than the difference in air-filled porosity, especially

at lower matric potentials. The air-filled porosity was approximately one-third higher in the middle layer than in the bottom layer, but the difference in gas diffusivity was less than 10 %. Gas diffusivity remained rather high even in nearly saturated peat





**Table 1.** Means and standard deviations for the measured air-filled porosity ($a$, m m$^{-3}$) of the peat samples at different matric potentials.

| Matric pot. | $-1$ kPa | | $-3$ kPa | | $-6$ kPa | | $-10$ kPa | |
|---|---|---|---|---|---|---|---|---|
| | $a$ | $n$ | $a$ | $n$ | $a$ | $n$ | $a$ | $n$ |
| 0–5 cm | 0.303±0.134 | 7 | 0.354±0.030 a | 4 | 0.385±0.077 a | 6 | 0.422±0.099 a | 5 |
| 20–25 cm | 0.119±0.044 a | 7 | 0.140±0.052 b | 4 | 0.203±0.061 b | 7 | 0.233±0.057 b | 6 |
| 40–45 cm | 0.031±0.020 b | 6 | 0.078±0.021 b | 4 | 0.135±0.045 b | 6 | 0.156±0.043 b | 6 |
| $p$ | 0.002 | | $< 0.001$ | | $< 0.001$ | | $< 0.001$ | |
| Test* | $t$ test | | ANOVA | | ANOVA | | ANOVA | |

*ANOVA: $F$ test and Tukey's test. ANOVA methods were not applicable to $-1$ kPa because of residual nonnormality and heteroscedasticity; an independent sample $t$ test was performed instead. Different letters indicate significant difference between depths ($p < 0.05$).

**Table 2.** Means and standard deviations for the measured $N_2$ diffusion coefficients ($D_s$, cm$^2$ s$^{-1}$) of the peat samples at different matric potentials.

| Matric pot. | $-1$ kPa | | $-3$ kPa | | $-6$ kPa | | $-10$ kPa | |
|---|---|---|---|---|---|---|---|---|
| | $D_s$ | $n$ | $D_s$ | $n$ | $D_s$ | $n$ | $D_s$ | $n$ |
| 0–5 cm | $(1.20\pm0.70)\times10^{-2}$ | 7 | $(1.06\pm0.22)\times10^{-2}$ a | 4 | $(1.55\pm0.57)\times10^{-2}$ a | 6 | $(1.92\pm0.40)\times10^{-2}$ a | 5 |
| 20–25 cm | $(3.64\pm1.26)\times10^{-3}$ a | 7 | $(5.29\pm0.85)\times10^{-3}$ b | 4 | $(5.56\pm2.01)\times10^{-3}$ b | 7 | $(8.60\pm3.40)\times10^{-3}$ b | 6 |
| 40–45 cm | $(2.83\pm0.96)\times10^{-3}$ a | 6 | $(4.75\pm1.97)\times10^{-3}$ b | 4 | $(5.73\pm3.76)\times10^{-3}$ b | 6 | $(8.29\pm4.11)\times10^{-3}$ b | 6 |
| $p$ | 0.26 | | 0.005 | | 0.001 | | 0.001 | |
| Test* | $t$ test | | ANOVA | | ANOVA | | ANOVA | |

*ANOVA: $F$ test and Tukey's test. ANOVA methods were not applicable to $-1$ kPa because of residual nonnormality and heteroscedasticity; an independent sample $t$ test was performed instead. Different letters indicate significant difference between depths ($p < 0.05$).

with low air-filled porosity. By contrast, the increase in gas diffusivity with increasing air-filled porosity was moderate under drier conditions. However, the number of measurements with an air-filled porosity greater than 0.4 m$^3$ m$^{-3}$ was rather small.

### 3.2 Pore network simulations of diffusion

The conditions at $-3$ kPa matric potential corresponded most closely to the configuration of the air-filled pore networks extracted from the µCT images, as the feature resolution of the images, 100 µm, roughly equals the minimum air-filled pore dimension at $-3$ kPa conditions. Because some of the gas diffusion measurements at $-3$ kPa were not successful, only four of the seven samples from each layer were applicable to the comparison. The porosities of the pore networks were generally very well in line with the air-filled porosities of the corresponding peat samples at $-3$ kPa conditions (Fig. 1a). The

difference between the measured air-filled porosity and the network porosity was not statistically significant (mean difference 0.01 m$^3$ m$^{-3}$; two-tailed paired sample $t$ test for log-transformed values, $t(11) = -0.472$ and $p = 0.65$) although it increased

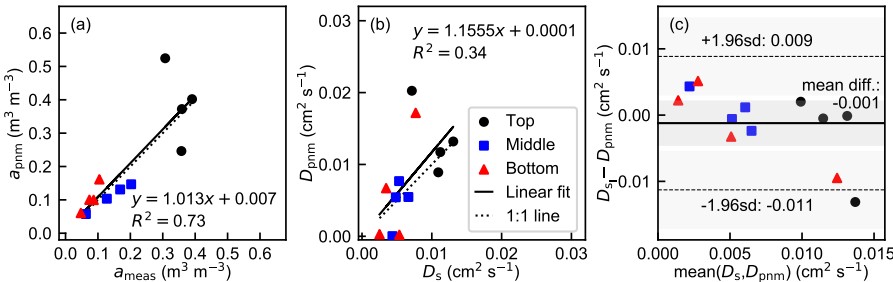

**Figure 1.** (a) Comparison of measurement-derived air-filled porosity ($a_{\mathrm{meas}}$) at $-3$ kPa matric potential and the air-filled porosity of corresponding pore networks ($a_{\mathrm{pnm}}$) for peat samples from the depths of 0–5 cm (top), 20–25 cm (middle), and 40–45 cm (bottom). The determination of $a_{\mathrm{meas}}$ was performed by subtracting the measured volumetric water content from the total porosity of the sample. (b) Comparison of soil gas diffusion coefficients determined from measurements ($D_{\mathrm{s}}$) at $-3$ kPa matric potential and obtained through pore network simulations ($D_{\mathrm{pnm}}$) for peat samples from different depths. (c) Bland–Altman plot showing the difference between the soil gas diffusion coefficients obtained with the two determination methods against the average of the values obtained by the methods for each sample. The solid line represents the mean difference, darker gray shading shows the 95 % confidence interval (CI) of the mean difference, and the dashed lines and the lighter gray shading show the 95 % limits of agreement and their 95 % CI, respectively.

with increasing porosity. The measured air-filled porosity was higher than the network porosity in all the bottom-layer samples while the opposite was the case in the middle layer. Network porosity was considerably higher than the measured air-filled porosity in one of the top-layer samples because the void fraction of the μCT image was considerably overestimated.

The soil gas diffusion coefficients obtained through the pore network simulations ($D_{\mathrm{pnm}}$) corresponded well to those determined from measurements (Fig. 1b). There was no significant difference between $D_{\mathrm{s}}$ and $D_{\mathrm{pnm}}$ (mean difference 0.001 cm$^2$ s$^{-1}$; two-tailed paired sample $t$ test, $t(11) = -0.783$ and $p = 0.45$). The values obtained with PNM were notably smaller than the measured values at the lower end of the range of $D$ (Fig. 1c). These cases corresponded to samples with low air-filled porosity, in which the connected pore networks extending through the whole sample domains were so sparse that their gas diffusion capability was very low. By contrast, two of the simulated soil gas diffusion coefficients were considerably higher than their measured counterparts. These discrepancies were due to overestimation of the network porosity (the top-layer sample) or a slight horizontal shrinkage generating continuous void space around the sample (the bottom-layer sample).

The pore network simulations showed considerable hysteresis in peat water content between wetting and drying conditions (Fig. 2). Air-filled porosity was higher at a specific matric potential during wetting than during drying, which resulted in a similar behavior in gas diffusivity. However, gas diffusivity in peat was lower at a specific air-filled porosity during wetting than during drainage (Fig. 2g–i).

### 3.3 Comparison of measured relative diffusivity with models

The agreement between measured relative diffusivities and those obtained with different gas diffusivity models varied with depth (Fig. 3). In the top layer with the highest porosity, the models tended to overestimate the relative diffusivity. By contrast,

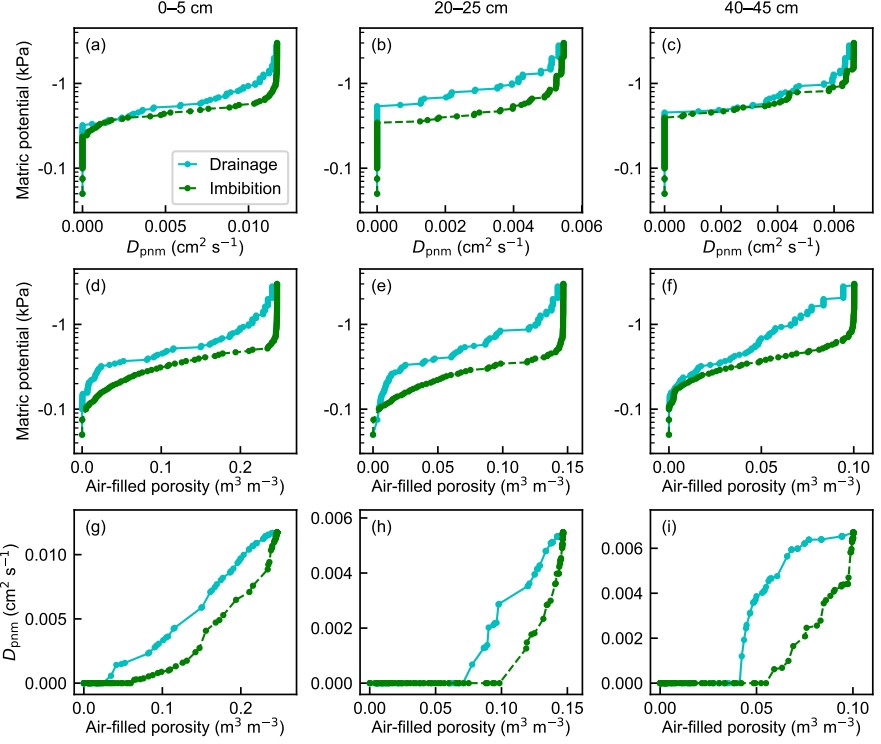

**Figure 2.** Simulation results for selected peat samples from (a,d,g) 0–5 cm, (b,e,h) 20–25 cm, and (c,f,i) 40–45 cm depths: (a–c) soil gas diffusion coefficient and (d–f) air-filled porosity as a function of matric potential and (g–i) soil gas diffusion coefficient as a function of air-filled porosity.

measured relative diffusivities were often considerably higher than any of the model-estimated values in the middle and bottom layers, especially at low air-filled porosity. The MQ61 model gave the best correspondence in the top layer, and the MQ60 model yielded the best agreement in the bottom layer (Table 3). In the middle layer the CC model performed best, although the TPM model and the CC model yielded quite similar results for the middle and bottom layers. However, the relative diffusivity of one of the middle-layer samples was low in comparison to air-filled porosity, which was reflected in the goodness-of-fit

analysis. If that sample was excluded from the analysis, the MQ60 model would give the best agreement followed by the TPM model and the CC model as in the bottom layer. Overall, the MQ61 model predicted the lowest values for all depths, whereas the MQ61 model gave the highest estimates.

The relative diffusivity at $-10$ kPa conditions as a function of air-filled porosity is used as a basis for the diffusivity parameterization in the TPM model. The measured relative diffusivities were generally higher than the estimate given by Eq.

(6) for air-filled porosity values less than 0.3, whereas the equation yielded substantially higher values than the measurements for a higher air-filled porosity (Fig. 4).





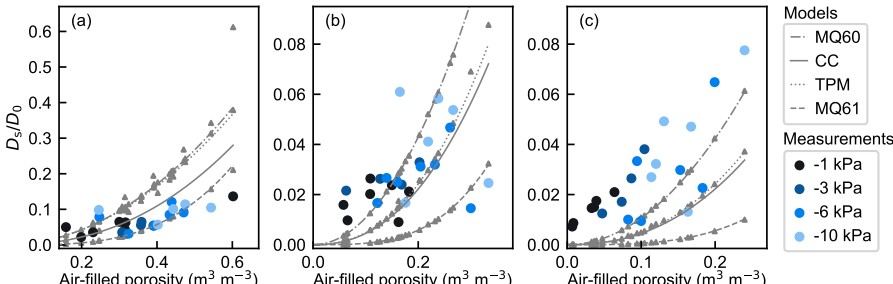

**Figure 3.** Measured (circular markers) and model-estimated (lines) relative diffusivity ($D_s/D_0$) against air-filled porosity for peat samples from the depths of (a) 0–5 cm, (b) 20–25 cm, and (c) 40–45 cm. Individual values of models MQ60, MQ61, and TPM are presented by triangular gray markers, and corresponding reference lines obtained using average porosity values for each depth are also shown.

**Table 3.** Lin's concordance correlation coefficients ($\rho_c$), coefficients of determination ($R^2$), and Akaike information criterion differences ($\Delta$AIC) for each of the applied models for relative diffusivity $D_s/D_0$.

|  |  | MQ61 | | | MQ60 | | | CC | | | TPM | |
|---|---|---|---|---|---|---|---|---|---|---|---|---|
|  | $n$ | $\rho_c$ | $R^2$ | $\Delta$AIC | $\rho_c$ | $R^2$ | $\Delta$AIC | $\rho_c$ | $R^2$ | $\Delta$AIC | $\rho_c$ | $R^2$ | $\Delta$AIC |
| 0–5 cm | 22 | 0.64 | $-0.44$ | 0 | 0.27 | $-10.6$ | 45.9 | 0.51 | $-2.04$ | 14.0 | 0.22 | $-19.6$ | 61.1 |
| 20–25 cm | 24 | 0.07 | $-2.49$ | 15.0 | 0.25 | $-3.44$ | 20.7 | 0.28 | $-1.07$ | 0 | 0.25 | $-1.55$ | 10.1 |
| 40–45 cm | 22 | 0.07 | $-1.76$ | 27.3 | 0.63 | 0.20 | 0 | 0.31 | $-0.63$ | 13.4 | 0.35 | $-0.51$ | 16.8 |

## 4 Discussion

### 4.1 Soil gas diffusivity measurements

The relative diffusivity values obtained in the study, 0.01–0.15, were within ranges previously reported for peat in the literature

(Iiyama and Hasegawa, 2005; Boon et al., 2013). Gas diffusivity was clearly highest in the top layer with the highest air-filled porosity, but the difference in gas diffusivity between the middle and bottom layers was rather small despite the notable decrease in air-filled porosity with depth (Table 2). Our findings are in line with a simulation study by Gharedaghloo et al. (2018) who found that gas diffusivity in *Sphagnum* peat decreased with depth at the topmost 15 cm layer. However, the gas diffusivities at the topmost 10 cm of the *Sphagnum* peat with an air-filled porosity range of 0.3 to 0.6 m³ m⁻³ were considerably higher than

our results for *Carex* peat from a drained peatland, which may reflect the differences in peat structure. The pore size distribution of *Carex* peat is known to differ from that of *Sphagnum* peat because of higher susceptibility to decomposition (McCarter et al., 2020), and enhanced decomposition due to oxic conditions in the unsaturated layer of a drained peatland further increases the differences (Kleimeier et al., 2017).

The results support the notion that macropore networks and their evolution with soil water content have a significant role in

gas diffusion in peat as suggested in Kiuru et al. (2022a). The gas diffusion capability of soil decreases with decreasing air-filled

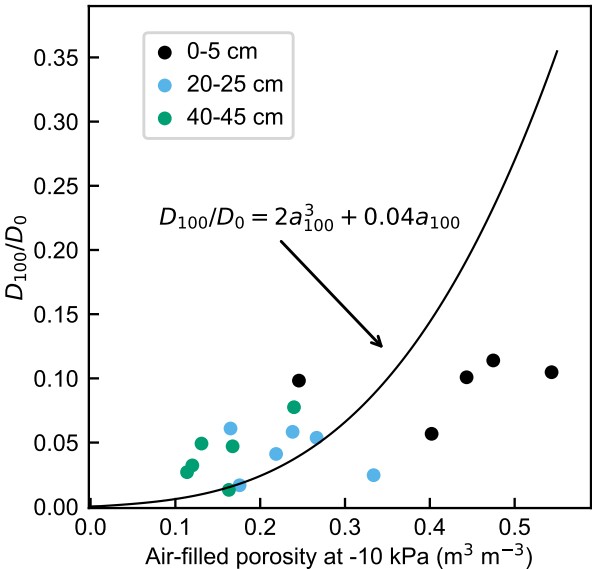

**Figure 4.** Measured relative diffusivity at $-100\,\mathrm{cmH_2O}$ ($-10\,\mathrm{kPa}$) matric potential as a function of air-filled porosity and the empirical equation by Moldrup et al. (2000).

porosity because of the diminishing and reshaping of the air-filled fraction of the pore network and the resulting decrease in pore connectivity (Moldrup et al., 2004). As smaller pores are filled with water, the air-filled pathways become more tortuous and the transport paths through the soil matrix become longer. In addition, pore dimensions and the total macropore volume are reduced with depth in peat because of an increasing degree of organic matter decomposition and higher compression by overlying matter (Rezanezhad et al., 2016). However, the fact that gas diffusivity was not extensively suppressed deeper in the peat profile despite the constant decrease of air-filled volume with depth indicates that the connectivity of the macropore network remains high enough for gas transport to be sustained even in low air-filled porosity conditions.

The wide range in the values of the soil gas diffusion coefficient between samples from the same depth revealed the large heterogeneity of peat structure (Fig. 3). The variability in gas diffusivity reflects not only the spatial variability in air-filled porosity but also the complexity of the geometry, dimensions, and connectivity of the macropore space (Kiuru et al., 2022a). In addition, gas diffusivity did not increase with porosity similarly in all samples because of the vertical variation in porosity in some samples. Vertical soil structure with alternating more porous and less porous layers may result in a fairly high average porosity but does not allow gas diffusion through the sample because of the obstructing effect induced by the less porous layers.

The shrinkage behavior of peat may have affected the soil gas diffusivity measurements under low-matric-potential conditions. Slight horizontal shrinkage was observed in many of the middle- and bottom-layer sample images, which were constructed at $-10\,\mathrm{kPa}$ conditions. The gradual development of shrinkage with drying may have resulted in a progressively overestimated measured soil gas diffusion coefficient under increasingly lower matric potentials because of the continuous





void space that was formed between the peat matrix and the cylinder walls. Gas leakage along the voids at the margin of the cylinder may have increased the observed gas diffusion rate.

The decrease in gas diffusivity with depth in wet conditions decreases the potential gas transfer rate through the unsaturated layer in peat. A simplified calculation for $O_2$ diffusivity illustrates the difference. Let an unsaturated zone of depth 0.5 m be divided into three discrete layers with equal thickness. Let us further assume that $O_2$ concentration at the surface corresponds to the atmospheric concentration of 300 g m$^{-3}$ and that the concentration at the WT level is zero. If there are no $O_2$ sinks in the unsaturated zone and the diffusivities of these layers equal the averages of our results with roughly corresponding

matric potential conditions (Table 2), the $O_2$ flux through the unsaturated zone is approximately 30 g m$^{-2}$ d$^{-1}$. If the soil gas diffusion coefficient is set equal to the top-layer value in the entire unsaturated layer, the flux is approximately 80 g m$^{-2}$ d$^{-1}$, that is, about 3 times as high. These kinds of totally constant conditions are obviously not realistic in nature because the vertical gradient of volumetric water content is high in unsaturated soil with a high WT, which strongly affects the change in soil gas diffusivity with depth.

## 4.2    Application of PNM to gas diffusion in peat

A network representation of pore space generated by X-ray tomography and image analysis provides detailed information on the topology and geometry of peat pore structures that cannot be obtained with traditional laboratory methods. PNM then combines the pore space connectivity described by network topology with a semianalytical description of transport processes between two neighboring pores (Blunt et al., 2013). The pore network structure and the dimensions of the flow routes govern

the capability of gas diffusion in peat. One of the aims of our study was to evaluate the applicability of PNM to the assessment of gas diffusivity in peat. Despite the limitations imposed by the applied imaging resolution (see Kiuru et al., 2022a) and the simplified geometry of the pore network model, the simulated soil gas diffusivity matched the measured values generally very well (Fig. 1). The smallest throat diameter distinguishable in the µCT images was 100 µm, which corresponds to a matric potential of about −3 kPa. Thus, the images have apparently contained sufficient information for a proper representation of

the pore structure relevant for gas diffusion at the −3 kPa conditions. The principal connected pore space made the major contribution to gas diffusion in the sample scale, whereas small throats and dead-end pores that were perhaps undetectable in the images may have had an insignificant effect on the gas transport behavior (Blunt et al., 2013). The pore network method is also suitable for the assessment of gas diffusivity in peat because the resolution of 100 µm is sufficient for an accurate characterization of gas diffusion in unsaturated peat, where the WT typically remains close to the surface and smaller pores are

generally filled with water.

However, not all simulation results corresponded to the measured values so closely. The primary reasons for the discrepancies were an incorrect pore network description induced by the limited imaging resolution and sample shrinkage. Pore network simulations underestimated the soil gas diffusion coefficient by an order of magnitude for some of the samples with an air-filled porosity of less than 0.1 m$^3$m$^{-3}$. The structure of these samples may have been so dense and the pore dimensions so

small that the narrowest portions of all the diffusion routes through the sample were largely indistinguishable in the images. By contrast, some of the soil gas diffusion coefficients were notably overestimated by the simulation. This resulted from an





incorrect description of the pore network geometry due to inaccurate estimation of the air-filled volume of the images or from an incorrect determination of the extent of the connected pore space due to sample shrinkage. Too high an intensity threshold used in the image solid–void classification stage increased the network volume and the pore and throat dimensions and thus
falsely enhanced the transport capacity of the network. Sample shrinkage affected the performance of the network generation process because the µCT imaging was performed for samples at $-10$ kPa conditions while the pore network distinguishable in the images corresponded to the extent of the air-filled pore space at $-3$ kPa conditions. If the sample had shrunk between $-3$ and $-10$ kPa conditions, the pore network structure extracted from the image was not representative for the conditions at $-3$ kPa. The shrinkage of the samples resulted in the generation of continuous void space between the sample and cylinder walls,
which was then classified as a part of the pore network and therefore increased the network transport capacity.

The applicability of PNM is strongly affected by the accuracy of the description of the pore space volume and dimensions by the network representation. The air-filled porosities of the networks corresponded very well to the measurement-based estimates at $-3$ kPa conditions (Fig. 1a). However, the inaccuracy of air-filled porosity calculations may complicate the issue. Values of peat particle density as low as 1300 kg m$^{-3}$ have been reported for peat (Päivänen, 1973). Using the value 1400
kg m$^{-3}$ instead of 1500 kg m$^{-3}$ for particle density would result in 0.005 to 0.008 m$^3$ m$^{-3}$ smaller air-filled porosity values within the bulk density range of the samples, 110 to 180 kg m$^{-3}$ as reported in (Kiuru et al., 2022a). That would affect the comparison especially at deeper layers with smaller air-filled porosity.

The effect of hysteresis on soil gas diffusivity is one of the issues that can be conveniently studied using PNM. Experimental determination of the differences in soil dynamics between drying and wetting conditions is complex and time-consuming (Likos
et al., 2014), whereas the consequences of hysteresis on the evolution of air-filled pore structure can readily be estimated through water retention and imbibition simulations. According to our results, gas diffusivity at a specific matric potential was higher during wetting than during drying, thus qualitatively following the behavior of air-filled porosity. However, gas diffusion was rapidly suppressed with decreasing air-filled porosity, while gas diffusivity increased considerably faster in drying conditions (Fig. 2). In addition, the threshold air-filled porosity for gas transport was substantially higher in wetting
than in drying conditions especially in deeper layers. This can be explained by the dynamics of pore filling and emptying. The largest pores were readily emptied of water when saturated soil started to dry, whereas the smallest pores were the first ones to be filled with water during wetting. When the smallest pores near the bottom of the sample started to fill with water with an increasing matric potential, gas diffusivity started to decrease quickly because the air-filled conduits between the top and bottom got blocked. The decrease was most pronounced in deeper layers where the fraction of the smallest pores was higher.
This diminished the quantitative effect of hysteresis on gas diffusivity in peat.

Upscaling issues are a common challenge in the application of µCT-related methods and in the comparison of pore network simulations with the results of measurements performed in core scale and thus in the assessment of the validity of the PNM approach (Blunt et al., 2013). High-resolution images of samples of size typically used in the measurements are very large, and image processing and pore network simulation in these networks would be computationally highly intensive. Therefore, the
domain size used in PNM is often considerably below the core scale (Gharedaghloo et al., 2018). The heterogeneous structure of peat complicates the upscaling from µCT scale to core scale because properties determined for a small volume may often not



be statistically representative for the whole sample. This is the case especially for transport properties, which are affected by the tortuosity and connectivity of the medium (Mostaghimi et al., 2013). However, we were able to simulate and reproduce the gas diffusion behavior in wet peat satisfactorily in the core scale by combining the simulation results for four parallel networks

despite the fact that some information on pore space connectivity was missed because of the domain division.

## 4.3 Applicability of gas diffusivity models to peat

Simple but efficient gas diffusivity models are needed for the characterization of gas transport in process-based models for simulating GHG production in peat and gas exchange between soil and the atmosphere (e.g., Fan et al., 2013; Raivonen et al., 2017) and for the estimation of soil GHG fluxes with the concentration gradient method (Sullivan et al., 2010; Maier and

Schack-Kirchner, 2014). Applicability of gas diffusivity models constructed for porous materials in general (for example, the models by Currie (1960), Millington and Quirk (1960), and Millington and Quirk (1961)) or for mineral soils (for example, the model by Moldrup et al. (2004)) to organic material such as peat is often considered poor (Iiyama and Hasegawa, 2005). None of the examined gas diffusivity models significantly outperformed the other models in our study. Higher model complexity did not increase its predictive capacity, as the three-parameter TPM model was not the best-performing option for any peat layer.

Our results are in line with Boon et al. (2013) who found that gas diffusion models, including CC and MQ61, underestimated the relative diffusivity in peat with air-filled porosity less than 0.2. However, relative diffusivity also remained higher than the model estimates under higher air-filled porosity conditions in Boon et al. (2013). Also, Hamamoto et al. (2016a) found that the MQ61 model underestimated relative diffusivity in peat especially at the conditions of low air-filled porosity. The theoretically based MQ61 model was derived assuming spherical pores and a uniform pore size distribution, which is a rough

simplification especially for peat that has a wide pore size distribution and where well-conducting macropores have a significant role. However, the MQ61 model has also been shown to overestimate relative diffusivity for high air-filled porosity values in mineral soils (Jin and Jury, 1996; Moldrup et al., 2000). Iiyama and Hasegawa (2005) found that the MQ61 model performed better than the TPM model, the latter of which overestimated relative diffusivity in peat. The TPM model was developed on the basis of gas diffusivity measurements in mineral soil, and therefore, it may have failed to account for the pore structure inherent

to peat soils. The application of gas diffusivity models with empirical parameters is known to be limited to soils similar to those used for calibration (Blagodatsky and Smith, 2012).

The models were not able to reproduce the observed gas diffusivity behavior under high-matric-potential conditions in peat. Measured relative diffusivities were high under the conditions of low air-filled porosity. This is in line with King and Smith (1987) who found that relative diffusivity in peat was generally higher at air-filled porosity values below $0.10 \mathrm{~m}^3 \mathrm{~m}^{-3}$ and

lower at air-filled porosity above $0.13 \mathrm{~m}^3 \mathrm{~m}^{-3}$ than corresponding literature values for mineral soils. A large number of natural macropores is a characteristic feature of peat (Lennartz and Liu, 2019). The macropores are formed by partially decomposed plant residues and therefore form a highly connected network functioning as a channel system that facilitates gas diffusion despite a low bulk air-filled porosity. By contrast, gas diffusivity did not increase so much with increasing air-filled porosity. The gas diffusivity of a porous medium is lower in the presence of more tortuous diffusion pathways (Moldrup et al., 2001).

This suggests that peat may display a more complex configuration of air-filled pores and a larger number of dead-end pores,



which do not contribute to gas transport, than mineral soil. Low gas diffusivity under the conditions of high air-filled porosity may also result from a vertically layered peat structure and anisotropic pore connectivity, which obstructs gas diffusion in the direction towards the atmosphere. In addition, the heterogeneous physical structure of peat impedes the determination of a specific gas diffusivity value for soil with certain bulk properties. Our results revealed large variation in gas diffusivity under

specific air-filled porosity conditions.

The rapid decrease in gas diffusivity with depth predicted by the gas diffusivity models has a remarkable effect on gas transfer capacity through the unsaturated layer in comparison to the measured soil gas diffusivity. This may result in a significant underestimation of gas transfer rates in peat. A simplified calculation for $CH_4$ diffusion shows the high impact of the difference between the measured and the model-estimated soil gas diffusion coefficients. Let an unsaturated zone of depth 0.5 m be

divided into three layers with equal thickness. Let us further set the $CH_4$ concentration in the air-filled pores at the WT level to 50 g m$^{-3}$ and the concentration at the surface to zero. If there are no $CH_4$ sinks in the unsaturated zone and the diffusivities of these layers equal the measured average values (Table 2) as in the example calculation in Sect. 4.1, the $CH_4$ flux through the unsaturated zone to the atmosphere is approximately 4 g m$^{-2}$ d$^{-1}$. By contrast, if the gas diffusivity model estimates calculated from the corresponding air-filled porosity values are used as the soil gas diffusion coefficients for each layer, the

flux is only 0.006 to 0.5 g m$^{-2}$ d$^{-1}$. The main reason for the small fluxes predicted by the gas diffusivity models is the small estimated soil gas diffusion coefficient in the bottommost layer with low air-filled porosity.

## 5   Conclusions

We studied gas diffusivity in unsaturated peat with laboratory experiments and pore network simulations. With some exceptions, the simulations conducted using the macropore networks constructed from the µCT images of the peat samples were

able to reproduce the measured gas diffusion dynamics characterized by the soil gas diffusion coefficient. Therefore, the µCT and PNM methods may offer a promising alternative to the traditional estimation of transport properties of peat through laboratory measurements, which are often prone to technical and procedural errors. The pore network approach also enables a more explicit investigation of relations between the physical structure of peat and its larger-scale gas transport properties. In addition, PNM can give insight into the pore-scale processes and phenomena that affect the GHG dynamics in peat. However,

the performance of PNM is strongly dependent on the quality of µCT imaging and image processing and the success of pore network generation. If peat pore space topology and pore geometry are correctly represented by the pore network object, our results indicate that soil gas diffusivity can be estimated adequately using the PNM approach.

Gas diffusivity measurements for peat assist in the choice of a suitable description of gas diffusion processes and a proper parameterization of the soil gas diffusion coefficient in process-based models that are used to simulate biogeochemical pro-

cesses and GHG production and emissions in peatlands. However, the gas diffusivity models investigated in this study were not very successful in estimating gas diffusivity in peat, especially in nearly saturated conditions. The gas diffusivity models constructed for uniform porous material or mineral soil were probably not able to account for the impact of the distinctive





structure of peat. This highlights the need for further experimental research on gas diffusivity in different types of unsaturated peat over a wide air-filled porosity range.

*Code and data availability.*    The data that support the findings of this study are available in GitHub [https://github.com/pjkiuru/macropore_diffusion] and at https://doi.org/10.5281/zenodo.6327112 (Kiuru et al., 2022b). The µCT image, binary image, and pore network data are available from the corresponding author upon reasonable request. The image processing and simulation parts of this study used the publicly available packages PoreSpy and OpenPNM. The Python scripts used in the calculations and the simulation output are available in GitHub [https://github.com/pjkiuru/macropore_diffusion].

*Author contributions.*    AL and TG developed the idea and designed the study. AL and MP collected the samples and performed the water retention and gas diffusivity measurements. LK performed the gas chromatography measurements. TG and MR organized the µCT imaging and 3D reconstruction. PK processed the images and designed and conducted the simulations. PK and AM conducted the computations and analyzed the data. PK performed the statistical analysis. PK wrote the manuscript with significant contributions from AM, AL, MP, and LK. All other authors provided edits and comments on the paper. AL and MR are responsible for the funding acquisition.

*Competing interests.*    The authors declare that they have no conflict of interest.

*Acknowledgements.*    This research has been supported by the Academy of Finland (grant nos. 325168 and 325169). LK holds a Marie Skłodowska-Curie Actions fellowship under the European Commission's Horizon 2020 program (grant no. 843511). MR acknowledges SRC at the Academy of Finland (SOMPA, no. 312932) and EU Horizon 2020 (VERIFY, no. 776810). This work used services of the Helsinki University X-Ray Micro-CT Laboratory, also funded by the Helsinki Institute of Life Science (HiLIFE) under the HAIP platform.





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
