# Peer review of "Pore network modeling as a new tool for determining gas diffusivity in peat"

_Biogeosciences, 2022_

## Referee Comment (RC1)

**General Comments:**

This study determined the gas diffusivity of forested peat and extracted the macropore networks from µCT images of peat to simulate the gas diffusion along the networks. The authors highlighted that the combination of the µCT and pore networking modeling provide reliable estimation of gas diffusivity of peat. This is a very interesting story because our understanding of the pore structure of fen peat as well as the gas diffusivity is indeed quite limited. The novelty of the manuscript is high and it is well written. But a moderate revision is necessary before publication:

My major concerns are:

1. The pore structure of peat is missing and the discussion is a little weak.

The authors analyzed the pore structure of peat but the pore features (e.g. tortuosity, pore connectivity, pore size distributions) are not shown and discussed. These parameters are quite important for gas diffusivity but were only mentioned in several sentences during discussion (e.g. line 413). I think the quality of the manuscript will be improved greatly if the authors could give solid discussion based on the parameters derived from peat soils. Maybe the authors have published this information previously, still a summary of these parameters is necessary.

2. The shrinkage information is unclear.

One unique property of peat is "shrink-swell", which is much greater than mineral soils (e.g. clay). This is also the reason that our understanding of the gas diffusivity of peat is limited. The authors indicated that the gas diffusion measurement may be affected by soil shrink. I think it is necessary to provide more detailed information on soil shrink: percent of volume change and at which pressures and for which soils the gas diffusion values are less affected by soil shrink.

**Specific comments:**

1. line 12, this conclusion is unclear. From table 1, for top soil (0-5 cm) the gas diffusivity is high at -1 kPa, but the air-filled porosity is also high 30 vol%. This is not near saturation. For deeper peat (40-45 cm), the gas diffusivity is low though it is more near saturation.

2. line 14, or you mean the traditional gas diffusivity models?

3. lines 37-39, "unsaturated" is a quite wide range. I would suggest using "when soil moisture is high".

4. lines 48-50, not true. Or you could refer to "natural peat" or "pristine peat". For degraded peat, the density as well as macroporosity decreases with depth.

5. line 267-268, this sentence is not necessary.

6. line 271, see above.

7. Figure 4, why select this pressure head, not others. At this pressure head, soil shrink occur and it affect the results the most (line 341). Also, this figure shows the only the top soils do not follow the function well. Because it has the highest macroporosity (42 vol%). This value is comparable to the total porosity of mineral soils. I think the authors could provide more detailed information to highlight the unique structure of peat.

8. lines 314-323, I think the authors could discuss a little more. Generally, peat types (fen, bogs) and decomposition/degradation stages (Liu and Lennartz, 2019) are two crucial factors affecting peat structure. I think the authors could compare the gas diffusivity for fens and bogs (from previous publications) at a comparable decomposition level (e.g. bulk density values).

9. Line 335, the pore structure (e.g. connectivity) of the samples should be provided.

10. lines 345-354, I do not understand why it is necessary to give this discussion. Or, just concise the paragraph.

11.  lines 388, Actually, you could estimate the pore size according to the capillary rise equation.

12. lines 389-392, Not so true. I mean, you determined the weight of the samples. The air-filled porosity could be justified according to the weight differences rather than using particle density or total porosity.

13. Section 4.3, I think the authors could explain the reasons together with highlighting the unique structure of peat.

14. Table 3. Please check the values again, especially for $R^2$.

**Reference:**

Liu, H., & Lennartz, B. (2019). Hydraulic Properties of Peat Soils along a Bulk Density Gradient– A Meta Study. Hydrological Processes, 33, 101–114.

---

## Author Comment (AC2)

**bg-2022-112**

**Author responses to comments of Referee #2**

We would like to thank the referee for the effort and time they put in to comment on our manuscript. We are grateful for their comments and will make every attempt to fully address these comments in the revised manuscript.

In the following list, the points raised by the referee are written in bold characters, whereas our responses are shown in regular characters.

Soil gas diffusivity and its controls are important issues in understanding greenhouse-gas dynamics of ecosystems. Knowledge gaps in this area are more obvious for organic soils than for mineral soils. Therefore the study of Kiuru et al. covers a relevant topic and principally aims and scopes of Biogeosciences. Although the study touches many several relevant aspects of the research area, nowhere a really satisfactory scientific depth is achieved. Therefore I suggest a complete revision of the manuscript with a stronger scientific focus on aspects that are well covered by the measurements and can be amended by a sufficient theoretical base.

We will revise the manuscript, extend the discussion, and highlight the connection between the experimental results and the theoretical framework and assumptions of the pore network modeling method on the basis of the comments and suggestions by both referees.

Introduction: Here I miss any review about specific soil-physical features of peats in contrast to mineral soils. The state of the art, especially considering water retention but in parts also for gas diffusion would allow to formulate specific hypotheses for the application of diffusion models.

As suggested, we will extend the Introduction to shortly summarize the specific physical properties of peat soils and to introduce the challenges in the application of the gas diffusivity models. In addition, we will revise the corresponding statements in the Discussion accordingly.

line 36: Consumption of O2 higher than the supply is not precise, you mean that diffusion cannot maintain O2 concentration above a critical level (which can be even zero in extreme cases).

We will clarify and revise the sentence as suggested.

line 52: This applies only when gas diffusion through the water is disregarded.

Earlier in the paragraph, we stated that the diffusion is significantly faster in air than in water, and therefore, we implicitly ignored the diffusion in the water phase in the following discussion in the paragraph. Also, totally isolated individual macropores or totally isolated macropore clusters (that is, with neither air-filled nor water-filled connections to the active pore network) do not contribute to any transport processes. We will remove the sentence for clarity.

Material and methods: Description of standard methods can be substantially shortened (by 50%)

We will condense the measurement descriptions in Sect. 2.2 somewhat. However, we want to keep a sufficient level of detail so that ambiguity can be prevented and the reproducibility of the measurements can be enabled.

line 87: Reference soil group Histosol, is it order from UsSoilTax or reference soil group from WRB? Anyway, classification should be done somewhat deeper (WRB principal qualifiers).

The soil classification follows the WRB classification system, and it was classified as Hemic Histosol. We will refine the classification.

line 104: Shrinkage has been measured, but not reported.

The results for vertical shrinkage are presented in Kiuru et al. (2022). The presence and amount of horizontal shrinkage at -10 kPa matric potential conditions could also be roughly estimated from the micro-computed tomography images. The shrinkage and its effect on diffusion properties was considered negligible at higher matric potentials. We will add information on the measured sample shrinkage and on the observations of shrinkage properties in the Results section.

line 119: I wonder about the extremely long closing times (60 and 120 minutes). With the very high Ds values this could let to high measuring errors.

Generally, the nitrogen mole fractions were well (on average approximately 8 percentage units) above the ambient value of 78 % during the second sampling about 120 minutes after closing the nitrogen tubes (the measurement data are available as stated in the "Code and data availability" section of the manuscript). Therefore, we consider the results very reliable in that sense. However, in the measurements with the highest diffusion coefficients, the top-layer samples at -10 kPa, the second samples were taken 119–123 minutes after closing the tubes. It is true that the chamber system may have been rather close to the equilibrium state during the time of the second sampling for the few samples with the highest diffusivity values. This may have resulted in too low a calculated gas diffusion coefficient. When looking at the concentrations of the second sampling, this may have been the case for three or four samples, one of which had already been discarded because of an inconsistent concentration value at the first sampling. However, we cannot definitely confirm this conclusion on the basis of the measurements because of the indefiniteness of the ambient nitrogen concentration value and other possible error sources in the measurements. We will mention the possible measurement error caused by the long measurement process time in the Discussion.

line 130: What was the exact criterion to discard measurements? How do you know that the other measurements were correct?

We compared (1) the measured nitrogen concentrations at the sampling times to the analytical concentration value (Bakker and Hidding, 1970) and (2) the measured nitrogen concentrations of the same sample between different matric potentials. The criterion (1) also compares the concentrations

of an individual pair of measurements between the two sampling times. The measurement was discarded if there was a very high discrepancy.

The temporal evolution of the chamber nitrogen concentration, C(t), is dependent on the geometry of the measurement system (which was unchanged during a measurement), the air-filled porosity a of the sample, and the soil gas diffusion coefficient D of the sample. The dependence on C(t) on a is much smaller than on D especially at higher t (approximately greater than 30 min for the scale of D and the system geometry in this study).

Figure AR1 shows an example of the measured nitrogen concentrations and analytical concentration curves for the bottom-layer samples at -3 kPa matric potential. The analytical curves are calculated using a single value for a and different values for D, and therefore, they are not totally accurate for different samples. We can see that the concentration values for the samples number 3 and 6 do not follow the analytical curves at all. Also, the measured concentration at the first sampling, 45-50 minutes after the start of the measurement, was notably lower than the corresponding concentration for the same sample at -6 kPa and -10 kPa matric potentials for samples number 3, 5, and 6. For example, the nitrogen mole fraction for sample number 5 was 0.882 at -3 kPa at 47 minutes, whereas it was 0.977 at -6 kPa at 49 minutes and 0.928 at -10 kPa at 68 minutes. Therefore, even though the concentrations for sample number 5 follow the analytical curve, the concentrations were not consistent with the concentrations at other matric potential conditions, and the measurement has therefore been incorrect. The calculated D would have been 0.013 cm2 s-1 at -3 kPa, whereas it was 0.0020 cm2 s-1 and 0.0054 cm2 s-1 at the increasingly lower matric potential conditions.

In this example, no striking discrepancies were seen for samples number 1, 2, 4, and 7, and thus these measurements can be considered much more reliable than those for samples number 3, 5, and 6 when looking at the consistency of the measured concentration values. Of course, there can be many other error sources in the measurements, but the rejection of a measurement based on concentration inconsistency is a very straightforward and justified procedure. We will extend the description of the criterion in the manuscript.

Figure AR1. Measured nitrogen mole fractions and analytical concentration curves for the bottomlayer samples at -3 kPa matric potential.

line 133: Why specific models have been chosen? The Millington-Kirk models have a mechanistic base with randomly disrupted capillary systems. The TPM-model is mostly empirical. Do you have ideas, which of these models are more sound to organic material?

We have selected the gas diffusivity models on the basis of their popularity and prior performance. The simpler models have been widely used in soil biogeochemical models and other gas flux estimates because of their simple structure, and the TPM model has been shown to have good performance for many soil types. The performance of the simpler models for peat has also been assessed in previous studies as discussed in Sect. 4.1, and therefore comparison to other studies became possible in our study. We consider that the reasoning for model selection is presented clearly in Sects. 2.3 and 4.1. We discuss the structure and applicability of the gas diffusivity models quite extensively in Sect. 4.3 and conclude that none of the models significantly outperformed the other ones for peat.

Is there a reason why CO2 has been measured?

The CO2 measurements were needed to account for respiration. We technically measured the diffusion of O2 into the chamber (while N2 was moving out of the chamber). However, some of this O2 was converted to CO2 on the way through the sample. So, effectively, we measured changes in the N2/(N2+O2+CO2) ratio rather than the actual N2 mole fraction. This implies that we assumed a respiratory quotient (that is, CO2 produced per O2 consumed) of 1. For brevity, however, we will not extend the already detailed measurement description in the manuscript.

line 175 following, pore network:

It is an impressive mathematical toolbox that has been used. However, I miss any critical view (this could be also in the discussion) about the validity of the completely artifical pore network model. Peats consist of fibres, clusters with a possibly strong anisotropy due to the good compressibility. Why the throat-bubble model should reflect the pore network? Are these assumptions robust or do they easily create biases in estimations? This can be checked by simulations (sensitivity analysis) of exemplary datasets.

The pore network model is not completely artificial because it is based on the real structure of the porous object obtained from micro-computed tomography images. The model would be completely artificial if it had been constructed using a synthetic random network that was tuned to match the observed effective diffusivity. The topology of the pore network reflects very closely to that of the pore space of the porous object, and the volumes and cross-sectional areas of the network elements (the volumes and areas of pores and the areas of throats) are well-defined. Only the shapes and therefore the diffusional conductance values of the elements are approximations through the stick-and-ball simplification in the pore network representation, and the segmentation of the pore space to individual pores is always based on subjectively established distinctions. In this case, the segmentation was based on the watershed algorithm of the network extraction tool.

The structural anisotropy of the porous medium is well accounted for in the topology of the pore network. The orientation of the transport conduits is represented by the orientation of the network

edges in the three-dimensional pore network. Also, the impact of the fibric structure on the properties of the air-filled macropore network of peat is rather small in wet conditions prevailing in our study.

The "validity" or the degree of performance of the pore network modeling methodology is assessed in this study by comparing the simulated diffusivity values to the measured values. By making the comparison we can see that the integrated effect or contribution of the network elements to the diffusion capability of the peat samples was generally well reproduced. We can therefore state that the effective performance of the simulation method was rather good.

In our opinion, a thorough sensitivity analysis of the geometrical structure of the well-founded and established methodology of pore network modeling is beyond the scope of this study. The study contains the generation and assessment of an experimental dataset of peat soil gas diffusivity and the assessment of the performance of the pore network modeling methodology considering the reproduction of the measurement results. We will revise the discussion by further assessing the applicability of the simplified pore geometry used in the gas diffusivity simulations with respect to the actual structure of peat.

**Results part:**

line 251 following: The database is weak and results do not contain surprising or interesting patterns, so strongly shorten!

There are few studies on gas diffusivity in peat (other studies are discussed in Sect. 4.1), and the sample sizes were generally very small in these studies. In comparison to these previous studies, our sample size is not very small. We consider that the experimental results presented for gas diffusivity in peat provide a relevant scientific contribution, and the assessment of the results is thus justified. The variability of peat properties due to its heterogeneous physical structure is also highlighted in the analysis of the results. However, we will slightly shorten the paragraph.

Shrinkage is a very critical issue but is completely disregarded in the measurements. I would suggest to correct epsilon for the reduced total volume.

The results for shrinkage are presented in Kiuru et al. (2022). The decrease in sample height was taken into account in the calculation of air-filled porosity and the soil gas diffusion coefficient at -10 kPa conditions. The description in Sect. 2.2 is therefore imprecise. We will add information on the measured sample shrinkage and on the observations of shrinkage properties in the Results section. We will also correct the text in Sect. 2.2 by mentioning the fact that the air-filled porosity was calculated from the reduced total volume at -10 kPa conditions.

**Table 1: two digits are sufficient**

We would prefer to present the air-filled porosity values with three significant figures as we have done for the volumetric water content and bulk density of the same samples in Kiuru et al. (2022) and for the soil gas diffusion coefficient in this manuscript.

**Table 2: "significant differences" between what?**

We meant that different letters in the table indicate a significant difference in the soil gas diffusion coefficient between different peat sampling depths. The same sentence structure is used in the footnote of Table 1, and we will clarify both occasions.

line 295: Hysteresis has been only modeled with a non-checked pore model with swellingshrinkage disregarded. This is no relevant scientific contribution.

The analysis of hysteresis through exactly similar pore network simulations was an important method in our earlier study on macropore networks of peat (Kiuru et al., 2022). In the study, we examined the evolution of the volume of connected air-filled peat pore space during both drying and wetting. In this manuscript, we have extended the assessment and discussion to gas diffusivity in peat using the same methodology. Thus, we consider this simulation method to be relevant. In addition, the shrinkage of the samples at the rather low maximum simulation pressure (high minimum matric potential) of 2.9 kPa can be considered rather small and thus ignored in the analysis.

**Discussion:**

I would expect a critical view on the scientific progress including some theoretical thoughts that support the rather weak empirical base.

We show that the simplified pore network representation of peat macropore space and the pore network simulations presented in the study are able to describe the gas diffusion properties of peat on the sample scale. We also state the benefits of the methods and the advances of this study in the Conclusions section.

As we state in the response to a previous comment ("line 175 following"), micro-computed tomography imaging reveals the real complex structure of the macropore space with a wide pore size distribution and an anisotropic nature. We also found that the simplified pore geometry applied was then sufficient to describe the effective gas diffusion behavior of the porous medium. We will highlight the connection between the results and the theoretical framework and the assumptions of pore network modeling and further discuss the applicability of the simplified pore geometry used in the simulations.

line 315 to 354: Is completely trivial and can be omitted, better is to check for critical issues in measurement quality.

Because there are few experimental studies available on gas diffusivity in peat, we think that it is justified to discuss the diffusivity measurements and to compare the results with other studies. In the section, we will extend the discussion on the variation of peat diffusivity between peat types. Following the recommendation of the other referee, we will also review our earlier results for the pore network connectivity of the peat samples, which strongly affects their diffusion capability. In addition, we will point out more issues concerning measurement discrepancies in the section, as we have described in the reply to a previous comment.

line 363: The problem is, that the pore network model does have a real theoretical foundation.

The theoretical foundation of the gas diffusion simulation, Fickian gas diffusion in tubes between pore centroids in an interconnected network, is well-established and validated in many previous studies concerning pore network modeling. The larger-scale connectivity of the pore space is captured by the pore network obtained through micro-computed tomography imaging, and the smaller-scale behavior is described semi-analytically in each network element (Blunt et al., 2013). The diameters of the pore network conduits in this study, 100 micrometers or higher, are well above the applicability threshold of Fickian diffusion. We will further justify the applicability of the pore network simulations as well as shortly discuss the theoretical foundation of the method in the Discussion.

**line 393: See above, hysteresis problem/shrinkage swelling**

We will add some discussion on the possible effect of peat volume changes on the hysteretic behavior. However, as stated above, we consider the shrinkage so low at higher-matric-potential conditions that its effect can be excluded from the simulations performed in this study.

**References**

Bakker, J. W. and Hidding, A.: The influence of soil structure and air content on gas diffusion in soils, Neth. J. Agric. Sci., 18, 37–48, https://doi.org/10.18174/njas.v18i1.17354, 1970.

Blunt, M. J., Bijeljic, B., Dong, H., Gharbi, O., Iglauer, S., Mostaghimi, P., Paluszny, A., and Pentland, C.: Pore-scale imaging and modelling, Adv. Water Resour., 51, 197–216, https://doi.org/10.1016/j.advwatres.2012.03.003, 2013.

Kiuru, P., Palviainen, M., Grönholm, T., Raivonen, M., Kohl, L., Gauci, V., Urzainki, I., and Laurén, A.: Peat macropore networks – new insights into episodic and hotspot methane emission, Biogeosciences, 19, 1959–1977, https://doi.org/10.5194/bg-19-1959-2022, 2022.

---

## Author Response (AR1)

**bg-2022-112**

**Author responses to Referee comments**

We would like to thank the referees for the effort and time they put in to comment on our manuscript. We are grateful for their comments and have made every attempt to fully address these comments in the revised manuscript.

In the following list, the points raised by the referees are written in bold characters, whereas our responses are shown in regular characters. The line numbers in the referee comments correspond to the line numbers in the original version of the manuscript, whereas the line numbers in our responses refer to the revised manuscript.

We have also corrected a typo in Sect. 2.1: the dimensions of the acrylic cylinder were 50 mm, not 50 cm.

L102–104: "Vertically oriented peat samples were extracted along the pit face into acrylic cylinders (diameter 50 mm, height 50 mm) using a sharp knife and scissors, paying attention to keeping the peat structure undisturbed."

**Referee #1**

**General Comments:**

This study determined the gas diffusivity of forested peat and extracted the macropore networks from  $\mu$ CT images of peat to simulate the gas diffusion along the networks. The authors highlighted that the combination of the  $\mu$ CT and pore networking modeling provide reliable estimation of gas diffusivity of peat. This is a very interesting story because our understanding of the pore structure of fen peat as well as the gas diffusivity is indeed quite limited. The novelty of the manuscript is high and it is well written. But a moderate revision is necessary before publication:

**My major concerns are:**

1. The pore structure of peat is missing and the discussion is a little weak.

The authors analyzed the pore structure of peat but the pore features (e.g. tortuosity, pore connectivity, pore size distributions) are not shown and discussed. These parameters are quite important for gas diffusivity but were only mentioned in several sentences during discussion (e.g. line 413). I think the quality of the manuscript will be improved greatly if the authors could give solid discussion based on the parameters derived from peat soils. Maybe the authors have published this information previously, still a summary of these parameters is necessary.

We agree that macropore network connectivity and other pore network features are significant issues in the gas diffusion capability of soil. In Kiuru et al. (2022), we have determined and presented network metrics for the pore structure of the same samples that were used in the gas diffusion measurements and simulations in this study. However, because of the shrinkage of the samples and computational limitations, connectivity metrics were calculated for subsamples, which were then assumed to be representative for the corresponding total sample volumes. Therefore, the network

metrics and the measured diffusivities are not directly comparable. We have summarized the results on macropore size and network connectivity metrics in Discussion (Sect. 4.1).

L349–359: "The structure and connectivity of the top-layer macropore networks differed greatly from those of the deeper-layer networks as the following short overview shows. The macropore network metrics are presented in detail in Kiuru et al. (2022). The average macropore volume in the top layer (0.32 mm3) was almost double the average volume in the middle and bottom layers (0.19 mm3 in both layers). However, the difference between median pore throat diameters was smaller, as the median diameters were 0.25, 0.23, and 0.20 mm in the top, middle, and bottom layer, respectively. The average pore coordination number, that is, the number of connections to a pore, was almost twice as large in the top-layer networks (6.0) as in the middle- and bottom-layer networks (3.3 and 3.1 in the middle and bottom layer, respectively). The geometrical tortuosity was also significantly lower in the top layer (1.6) than in the deeper layers (2.7 and 2.3). Thus, the lower connectivity and higher tortuosity of the middle-layer and bottom-layer samples were reflected in a lower gas diffusivity. However, the network metrics are not directly comparable to the diffusion measurements because the volume of the network domain used in the calculation of the network metrics was smaller than the total sample volume."

**2. The shrinkage information is unclear.**

One unique property of peat is "shrink-swell", which is much greater than mineral soils (e.g. clay). This is also the reason that our understanding of the gas diffusivity of peat is limited. The authors indicated that the gas diffusion measurement may be affected by soil shrink. I think it is necessary to provide more detailed information on soil shrink: percent of volume change and at which pressures and for which soils the gas diffusion values are less affected by soil shrink.

The vertical shrinkage of the samples was measured after the diffusion experiment (at -10 kPa matric potential conditions), and the results are presented in Kiuru et al. (2022). The presence and amount of horizontal shrinkage could be roughly estimated from the micro-computed tomography images. The shrinkage and its effect on diffusion properties was considered negligible at higher matric potentials. We have added information on the measured sample shrinkage and on the observations of shrinkage properties in the Results section. We have also mentioned the fact that the soil gas diffusion coefficient was calculated using the reduced sample height at -10 kPa conditions.

L268–271: "The shrinkage of the samples in the vertical direction at -10 kPa matric potential was on average 6.3 % in the top layer, 3.7 % in the middle layer, and 2.3 % in the bottom layer. The vertical shrinkage was considered to be negligible at higher-matric-potential conditions. In addition, slight horizontal shrinkage was observed in some of the middle-layer and bottom-layer samples at -10 kPa conditions. The estimated decrease in the sample diameter was generally of the order of 1 to 2 % or less."

L138–139: "The decrease in the sample length due to shrinkage was taken into account in the calculation at -10 kPa conditions."

Specific comments:

1. line 12, this conclusion is unclear. From table 1, for top soil (0-5 cm) the gas diffusivity is high at -1 kPa, but the air-filled porosity is also high 30 vol%. This is not near saturation. For deeper peat (40-45 cm), the gas diffusivity is low though it is more near saturation.

The measured relative diffusivity values were greater than 0.01, mostly greater than 0.02, at air-filled porosity less than 0.1 m3 m-3. Measurements of gas diffusion in mineral soil have given lower values for relative diffusivity near saturation (Moldrup et al., 2004), and all the studied gas diffusivity models predicted a relative diffusivity of less than 0.01 at an air-filled porosity less than 0.1 m3 m-3. Therefore, the measured gas diffusivity was higher than the diffusivity model predictions and higher than that of many mineral soils, so we conclude that the diffusivity was relatively high compared to mineral soils or model predictions and that it was not extremely low even in wet conditions. We have clarified the wording in the text.

L12–14: "However, gas diffusivity was not extremely low close to saturation, which may indicate that the structure of the macropore network is such that it enables the presence of connected diffusion pathways through the peat matrix even in wet conditions."

2. line 14, or you mean the traditional gas diffusivity models?

Yes, we mean the traditional or conventional, commonly used gas diffusivity models mentioned in line 11. We have changed the wording.

L14–17: "The traditional gas diffusivity models were not very successful in predicting the soil gas diffusion coefficient. This may indicate that the microstructure of peat differs considerably from the structure of mineral soils and other kinds of porous materials, for which these models have been constructed and calibrated."

3. lines 37-39, "unsaturated" is a quite wide range. I would suggest using "when soil moisture is high".

Because the water table is rather close to the soil surface in undrained peatlands, it is often assumed that soil moisture is high in unsaturated peat, and therefore, it is not considered necessary to refer to high-moisture unsaturated peat in particular in the literature. However, near-surface peat may occasionally be drier in drained peatlands. We have clarified the wording as suggested.

L37–39: "This is the case below the WT, where anaerobic conditions dominate permanently (Abdalla et al., 2016), but anaerobic conditions may also occur in high-moisture unsaturated peat if the peat structure does not favor  $O_2$  transport (Fan et al., 2014)."

4. lines 48-50, not true. Or you could refer to "natural peat" or "pristine peat". For degraded peat, the density as well as macroporosity decreases with depth.

We have clarified the sentence and moved it to the succeeding paragraph.

L57–59: "Peat macroporosity is generally highest near the surface in undrained peatlands because the degree of decomposition typically increases with depth and because the decomposition results in decreasing pore volumes (Päivänen, 1973)."

5. line 267-268, this sentence is not necessary.

6. line 271, see above.

We have removed the sentences as suggested.

7. Figure 4, why select this pressure head, not others. At this pressure head, soil shrink occur and it affect the results the most (line 341). Also, this figure shows the only the top soils do not follow the function well. Because it has the highest macroporosity (42 vol%). This value is comparable to the total porosity of mineral soils. I think the authors could provide more detailed information to highlight the unique structure of peat.

As we state in Sect. 3.3, the TPM model uses the relative diffusivity at -10 kPa conditions as the basis for the gas diffusivity parameterization. Therefore, we illustrate the behavior of gas diffusivity at that matric potential in the figure. As suggested, we have extended the discussion on the performance of the TPM model parameterization in Sect. 4.3 in the Discussion.

L499–503: "By contrast, the measured gas diffusivity did not increase with increasing air-filled porosity as much as the gas diffusivity models estimated. For example, the parameterization (Eq. 6) used in the TPM model failed to capture the measured relative diffusivity at the upper end of the air-filled porosity range, 0.40–0.55 m3 m-3 (Fig. 4). These values are close to the total porosity of many mineral soils (Hillel, 1998). The TPM model, being developed on the basis of gas diffusivity measurements in mineral soil, may have failed to account for the pore structure and high macroporosity inherent to peat soils."

8. lines 314-323, I think the authors could discuss a little more. Generally, peat types (fen, bogs) and decomposition/degradation stages (Liu and Lennartz, 2019) are two crucial factors affecting peat structure. I think the authors could compare the gas diffusivity for fens and bogs (from previous publications) at a comparable decomposition level (e.g. bulk density values).

Unfortunately, studies on gas diffusivity in peat are very scarce especially for fens, and the sample size of most of the studies is small. Therefore, it is difficult to perform a more detailed comparison between studies or between fens and bogs. The available studies have been listed in the discussion. We have extended the discussion on the basis of available literature.

L331–339: "However, the gas diffusivities at the topmost 10 cm of the Sphagnum peat with an airfilled porosity range of 0.3 to 0.6 m3 m-3 were considerably higher than our results for Carex peat from a drained peatland, which may reflect the differences in peat structure. Peat type and decomposition stage are important factors affecting peat structure and transport properties (Liu and Lennartz, 2019). The pore size distribution of Carex peat is known to differ from that of Sphagnum peat because of higher susceptibility to decomposition (McCarter et al., 2020), and enhanced decomposition due to oxic conditions in the unsaturated layer of a drained peatland further increases the differences (Kleimeier et al., 2017). By contrast, the relative diffusivity of slightly or moderately decomposed Sphagnum-dominated peat with an air-filled porosity of less than 0.15 m3 -3 was generally lower than 0.02 and often lower than 0.01 in liyama and Hasegawa (2005) and in Hamamoto et al. (2016a), whereas the relative diffusivity was always 0.01 or higher and sometimes as high as 0.04 under similar conditions in our study."

9. Line 335, the pore structure (e.g. connectivity) of the samples should be provided.

In Kiuru et al. (2022), we have determined and presented metrics for the pore structure and connectivity of the same samples that were used in the gas diffusion measurements and simulations in this study. However, because of the shrinkage of the samples and computational limitations, the connectivity metrics were calculated for subsamples, which were then assumed to be representative for the corresponding total sample volumes. We have given an overview of the network metrics in Discussion and an example of network connectivity metrics between samples from a single depth in this paragraph.

L349–359: "The structure and connectivity of the top-layer macropore networks differed greatly from those of the deeper-layer networks as the following short overview shows. The macropore network metrics are presented in detail in Kiuru et al. (2022). The average macropore volume in the top layer (0.32 mm3) was almost double the average volume in the middle and bottom layers (0.19 mm3 in both layers). However, the difference between median pore throat diameters was smaller, as the median diameters were 0.25, 0.23, and 0.20 mm in the top, middle, and bottom layer, respectively. The average pore coordination number, that is, the number of connections to a pore, was almost twice as large in the top-layer networks (6.0) as in the middle- and bottom-layer networks (3.3 and 3.1 in the middle and bottom layer, respectively). The geometrical tortuosity was also significantly lower in the top layer (1.6) than in the deeper layers (2.7 and 2.3). Thus, the lower connectivity and higher tortuosity of the middle-layer and bottom-layer samples were reflected in a lower gas diffusivity. However, the network metrics are not directly comparable to the diffusion measurements because the volume of the network domain used in the calculation of the network metrics was smaller than the total sample volume."

L360–364: "The wide range in the values of the soil gas diffusion coefficient between samples from the same depth revealed the large heterogeneity of peat structure (Fig. 3). The variability in gas diffusivity reflects not only the spatial variability in air-filled porosity but also the complexity of the geometry, dimensions, and connectivity of the macropore space (Kiuru et al., 2022). For example, the average pore coordination number varied from 3.8 to 8.3 and the geometrical tortuosity from 1.4 to 1.8 in the top-layer networks."

10. lines 345-354, I do not understand why it is necessary to give this discussion. Or, just concise the paragraph.

In the paragraph, we want to point out that a decrease in the gas diffusivity of peat with depth strongly affects the potential gas transfer rate in the soil. Further, we want to emphasize that it is important to take the vertical variation of gas diffusivity into account in, for example, the estimates of soil aeration rate, atmosphere–soil carbon exchange, and soil carbon budget. The calculation presented in the paragraph gives an example of the impact of the diffusivity variation on soil aeration rate, which further affects, for example, the rate of carbon degradation processes in the unsaturated layer. We have clarified the text and justified the relevance of the discussion.

L377–379: "The decrease in gas diffusivity with depth in wet conditions decreases the potential gas transfer rate through the unsaturated layer in peat. Therefore, it should be taken into account in, for example, the estimates of soil aeration and the development of process-based biogeochemical models."

11. lines 388, Actually, you could estimate the pore size according to the capillary rise equation.

We have determined, analyzed, and presented the pore sizes of the samples in Kiuru et al. (2022). The pore dimensions were determined from the micro-computed tomography images. The pore size distribution obtained from the water retention curve and capillary rise equation is more inaccurate than the image-based method because it gives the diameters of pore openings (throats) rather than pore volumes or pore body diameters and because it relies on the dynamic accessibility of pores to the invading fluid (Nimmo, 2005). Entrance of air to larger pores near the bottom of the sample can be prevented by the existence of smaller pores above them. Also, both the diameters of the pore openings and the cross-sectional diameters of the pore bodies are used in the simulations based on the pore network geometry and dimensions. The diffusional conductance between two pores is affected by the diameter of both pores as well as the diameter of the throat between them.

12. lines 389-392, Not so true. I mean, you determined the weight of the samples. The air-filled porosity could be justified according to the weight differences rather than using particle density or total porosity.

The relative volume of air as a fraction of the total pore space volume ("the degree of air saturation") can be determined from the weight differences between the samples if it is assumed that there is no air present in the sample in the saturated state. However, air-filled porosity is defined as the air volume present in the soil relative to the total soil volume, not to the total pore space volume. In addition, the air-filled porosity is usually greater than zero even in the saturated sample, and therefore, the total porosity of the sample must be calculated using the estimated particle density of the sample.

13. Section 4.3, I think the authors could explain the reasons together with highlighting the unique structure of peat.

As suggested, we have restructured Sect. 4.3, separating the comparisons to previous studies from the analysis of the reasons behind the observed mismatch of the gas diffusivity models.

14. Table 3. Please check the values again, especially for R2.

The determination of the performance statistic was ambiguous in the manuscript. The values for R2 were calculated with Eq. (9) from the residuals between measured and model-estimated values and the differences between the measured values and their mean. When comparing the measured values directly to the model-estimated values instead of the linear regression between the measured and model-estimated values, it is more correct to refer to this statistic as to the Nash–Sutcliffe efficiency (Moriasi et al., 2007). In this case, the sum of squares of the model error may be greater than the sum of squares of the differences between the measurements and their mean, and therefore the value of the statistic can be negative. We have changed the term for the statistic from the coefficient of determination to the Nash–Sutcliffe efficiency in Sect. 2.6 and in Table 3.

L240: "The Nash–Sutcliffe efficiency (Moriasi et al., 2007) is defined as [...]"

Reference:

Liu, H., & Lennartz, B. (2019). Hydraulic Properties of Peat Soils along a Bulk Density Gradient– A Meta Study. Hydrological Processes, 33, 101–114.

**References**

Kiuru, P., Palviainen, M., Grönholm, T., Raivonen, M., Kohl, L., Gauci, V., Urzainki, I., and Laurén, A.: Peat macropore networks – new insights into episodic and hotspot methane emission, Biogeosciences, 19, 1959–1977, https://doi.org/10.5194/bg-19-1959-2022, 2022.

Moldrup, P., Olesen, T., Yoshikawa, S., Komatsu, T., and Rolston, D. E.: Three-porosity model for predicting the gas diffusion coefficient in undisturbed soil, Soil Sci. Soc. Am. J., 68, 750–759, https://doi.org/10.2136/sssaj2004.7500, 2004.

Moriasi, D. N., Arnold, J. G., Liew, M. W. V., Bingner, R. L., Harmel, R. D., and Veith, T. L.: Model evaluation guidelines for systematic quantification of accuracy in watershed simulations, Trans. ASABE, 50, 885–900, https://doi.org/10.13031/2013.23153, 2007.

Nimmo, J. R.: Porosity and pore-size distribution, in: Encyclopedia of Soils in the Environment, edited by Hillel, D., pp. 295–303, Elsevier, Oxford, UK, ISBN: 978-0-12-348530-4, 2005.

**Referee #2**

Soil gas diffusivity and its controls are important issues in understanding greenhouse-gas dynamics of ecosystems. Knowledge gaps in this area are more obvious for organic soils than for mineral soils. Therefore the study of Kiuru et al. covers a relevant topic and principally aims and scopes of Biogeosciences. Although the study touches many several relevant aspects of the research area, nowhere a really satisfactory scientific depth is achieved. Therefore I suggest a complete revision of the manuscript with a stronger scientific focus on aspects that are well covered by the measurements and can be amended by a sufficient theoretical base.

We have revised the manuscript, extended the discussion, and highlighted the connection between the experimental results and the theoretical framework and assumptions of the pore network modeling method on the basis of the comments and suggestions by both referees.

Introduction: Here I miss any review about specific soil-physical features of peats in contrast to mineral soils. The state of the art, especially considering water retention but in parts also for gas diffusion would allow to formulate specific hypotheses for the application of diffusion models.

As suggested, we have extended the Introduction to shortly summarize the specific physical properties of peat soils and to introduce the challenges in the application of the gas diffusivity models. In addition, we have revised the corresponding statements in the Discussion accordingly.

L54–62: "The pore structure and other physical characteristics of peat differ considerably from those of mineral soil (McCarter et al., 2020). The unique properties such as high total porosity and low bulk density as well as the tendency to shrink and swell during drying and wetting affect the gas transport properties of peat (Rezanezhad et al., 2016). Peat pore space is characterized by macropores between partially decomposed plant remains and smaller pores inside the remains (Weber et al., 2017). Peat macroporosity is generally highest near the surface in undrained peatlands because the degree of decomposition typically increases with depth and because the decomposition results in decreasing pore volumes (Päivänen, 1973). Because of the high total porosity, the water retention capacity of peat is higher than that of mineral soils especially in wet conditions (Paavilainen and Päivänen, 1995, Walczak et al., 2002). The macropore network is considered more complex and more tortuous in peat than in mineral soils, which may result in a lower diffusion capability (liyama and Hasegawa, 2005)."

L68–72: "Gas diffusivity models are needed in process-based models describing biogeochemical processes and GHG production and emission in soils (Blagodatsky and Smith, 2012; Xu et al., 2016). However, the applicability of the models to peat is questionable because of the unique physical characteristics of peat (liyama and Hasegawa, 2005). The impact of the complex and tortuous pore structure of peat may not be adequately represented in models designed for simplified porous media or mineral soils."

L504–506: "This supports the notion that peat may display a more complex configuration of air-filled pores and a larger number of dead-end pores, which do not contribute to gas transport, than mineral soil."

line 36: Consumption of O2 higher than the supply is not precise, you mean that diffusion cannot maintain O2 concentration above a critical level (which can be even zero in extreme cases).

We have clarified and revised the sentence as suggested.

L34–37: " $O_2$  is transported from the atmosphere into the peat where it is continuously consumed by heterotrophic respiration, which produces  $CO_2$ . If the rate of  $O_2$  consumption exceeds its supply,  $O_2$  may become depleted in the peat. Under these conditions, the microbial metabolism changes to other electron acceptors, and finally the production of  $CH_4$  starts (Bridgham et al., 2013)."

line 52: This applies only when gas diffusion through the water is disregarded.

Earlier in the paragraph, we stated that the diffusion is significantly faster in air than in water, and therefore, we implicitly ignored the diffusion in the water phase in the following discussion in the paragraph. Also, totally isolated individual macropores or totally isolated macropore clusters (that is, with neither air-filled nor water-filled connections to the active pore network) do not contribute to any transport processes. We have removed the sentence for clarity.

Material and methods: Description of standard methods can be substantially shortened (by 50%)

We have condensed the measurement descriptions in Sect. 2.2 somewhat. However, we want to keep a sufficient level of detail so that ambiguity can be prevented and the reproducibility of the measurements can be enabled.

line 87: Reference soil group Histosol, is it order from UsSoilTax or reference soil group from WRB? Anyway, classification should be done somewhat deeper (WRB principal qualifiers).

The soil classification follows the WRB classification system, and it was classified as Hemic Histosol. We have refined the classification.

L94: "The soil type is Hemic Histosol and the peat type is Carex peat."

line 104: Shrinkage has been measured, but not reported.

The results for vertical shrinkage are presented in Kiuru et al. (2022). The presence and amount of horizontal shrinkage at -10 kPa matric potential conditions could also be roughly estimated from the micro-computed tomography images. The shrinkage and its effect on diffusion properties was considered negligible at higher matric potentials. We have added information on the measured sample shrinkage and on the observations of shrinkage properties in the Results section.

L268–271: "The shrinkage of the samples in the vertical direction at –10 kPa matric potential was on average 6.3 % in the top layer, 3.7 % in the middle layer, and 2.3 % in the bottom layer. The vertical

shrinkage was considered to be negligible at higher-matric-potential conditions. In addition, slight horizontal shrinkage was observed in some of the middle-layer and bottom-layer samples at –10 kPa conditions. The estimated decrease in the sample diameter was generally of the order of 1 to 2 % or less."

line 119: I wonder about the extremely long closing times (60 and 120 minutes). With the very high Ds values this could let to high measuring errors.

Generally, the nitrogen mole fractions were well (on average approximately 8 percentage units) above the ambient value of 78 % during the second sampling about 120 minutes after closing the nitrogen tubes (the measurement data are available as stated in the "Code and data availability" section of the manuscript). Therefore, we consider the results very reliable in that sense. However, in the measurements with the highest diffusion coefficients, the top-layer samples at -10 kPa, the second samples were taken 119–123 minutes after closing the tubes. It is true that the chamber system may have been rather close to the equilibrium state during the time of the second sampling for the few samples with the highest diffusivity values. This may have resulted in too low a calculated gas diffusion coefficient. When looking at the concentrations of the second sampling, this may have been the case for three or four samples, one of which had already been discarded because of an inconsistent concentration value at the first sampling. However, we cannot definitely confirm this conclusion on the basis of the measurements because of the indefiniteness of the ambient nitrogen concentration value and other possible error sources in the measurements. We have mentioned the possible measurement error caused by the long measurement process time in the Discussion.

L372–376: "Gas leakage along the voids at the margin of the cylinder may have increased the observed gas diffusion rate. By contrast, the estimated  $N_2$  concentration difference between the diffusion chamber and free air was rather small during the second gas sampling in some of the top-layer samples with a high air-filled porosity at -10 kPa conditions, and the system may therefore have been close to equilibrium by that time. This may have resulted in too low a calculated soil gas diffusion coefficient."

line 130: What was the exact criterion to discard measurements? How do you know that the other measurements were correct?

We compared (1) the measured nitrogen concentrations at the sampling times to the analytical concentration value (Bakker and Hidding, 1970) and (2) the measured nitrogen concentrations of the same sample between different matric potentials. The criterion (1) also compares the concentrations of an individual pair of measurements between the two sampling times. The measurement was discarded if there was a very high discrepancy.

The temporal evolution of the chamber nitrogen concentration, C(t), is dependent on the geometry of the measurement system (which was unchanged during a measurement), the air-filled porosity a of the sample, and the soil gas diffusion coefficient D of the sample. The dependence on C(t) on a is much smaller than on D especially at higher t (approximately greater than 30 min for the scale of D and the system geometry in this study).

Figure AR1 shows an example of the measured nitrogen concentrations and analytical concentration curves for the bottom-layer samples at -3 kPa matric potential. The analytical curves are calculated using a single value for a and different values for D, and therefore, they are not totally accurate for different samples. We can see that the concentration values for the samples number 3 and 6 do not follow the analytical curves at all. Also, the measured concentration at the first sampling, 45–50 minutes after the start of the measurement, was notably lower than the corresponding concentration for the same sample at -6 kPa and -10 kPa matric potentials for samples number 3, 5, and 6. For example, the nitrogen mole fraction for sample number 5 was 0.882 at -3 kPa at 47 minutes, whereas it was 0.977 at -6 kPa at 49 minutes and 0.928 at -10 kPa at 68 minutes. Therefore, even though the concentrations for sample number 5 follow the analytical curve, the concentrations were not consistent with the concentrations at other matric potential conditions, and the measurement has therefore been incorrect. The calculated D would have been 0.013 cm2 s-1 at -3 kPa, whereas it was 0.0020 cm2 s-1 and 0.0054 cm2 s-1 at the increasingly lower matric potential conditions.

In this example, no striking discrepancies were seen for samples number 1, 2, 4, and 7, and thus these measurements can be considered much more reliable than those for samples number 3, 5, and 6 when looking at the consistency of the measured concentration values. Of course, there can be many other error sources in the measurements, but the rejection of a measurement based on concentration inconsistency is a very straightforward and justified procedure. We have extended the description of the criterion in the manuscript.

L139–142: "Of a total of 84 diffusion measurements, 16 were discarded because of strikingly inconsistent N2 concentration values for a sample between the gas sampling times or for a sample between different matric potential conditions at corresponding sampling times. The inconsistency was most probably caused by leakages in the diffusion measurement system or during gas sampling."

line 133: Why specific models have been chosen? The Millington-Kirk models have a mechanistic base with randomly disrupted capillary systems. The TPM-model is mostly empirical. Do you have ideas, which of these models are more sound to organic material?

We have selected the gas diffusivity models on the basis of their popularity and prior performance. The simpler models have been widely used in soil biogeochemical models and other gas flux estimates because of their simple structure, and the TPM model has been shown to have good performance for many soil types. The performance of the simpler models for peat has also been assessed in previous studies as discussed in Sect. 4.1, and therefore comparison to other studies became possible in our study. We consider that the reasoning for model selection is presented clearly in Sects. 2.3 and 4.1. We discuss the structure and applicability of the gas diffusivity models quite extensively in Sect. 4.3 and conclude that none of the models significantly outperformed the other ones for peat.

**Is there a reason why CO2 has been measured?**

The CO2 measurements were needed to account for respiration. We technically measured the diffusion of O2 into the chamber (while N2 was moving out of the chamber). However, some of this O2 was converted to CO2 on the way through the sample. So, effectively, we measured changes in the N2/(N2+O2+CO2) ratio rather than the actual N2 mole fraction. This implies that we assumed a respiratory quotient (that is, CO2 produced per O2 consumed) of 1. For brevity, however, we have not extended the already detailed measurement description in the manuscript.

**line 175 following, pore network:**

It is an impressive mathematical toolbox that has been used. However, I miss any critical view (this could be also in the discussion) about the validity of the completely artifical pore network model. Peats consist of fibres, clusters with a possibly strong anisotropy due to the good compressibility. Why the throat-bubble model should reflect the pore network? Are these assumptions robust or do they easily create biases in estimations? This can be checked by simulations (sensitivity analysis) of exemplary datasets.

The pore network model is not completely artificial because it is based on the real structure of the porous object obtained from micro-computed tomography images. The model would be completely artificial if it had been constructed using a synthetic random network that was tuned to match the observed effective diffusivity. The topology of the pore network reflects very closely to that of the pore space of the porous object, and the volumes and cross-sectional areas of the network elements (the volumes and areas of pores and the areas of throats) are well-defined. Only the shapes and therefore the diffusional conductance values of the elements are approximations through the stick-and-ball simplification in the pore network representation, and the segmentation of the pore space to individual pores is always based on subjectively established distinctions. In this case, the segmentation was based on the watershed algorithm of the network extraction tool.

The structural anisotropy of the porous medium is well accounted for in the topology of the pore network. The orientation of the transport conduits is represented by the orientation of the network edges in the three-dimensional pore network. Also, the impact of the fibric structure on the properties of the air-filled macropore network of peat is rather small in wet conditions prevailing in our study.

The "validity" or the degree of performance of the pore network modeling methodology is assessed in this study by comparing the simulated diffusivity values to the measured values. By making the comparison we can see that the integrated effect or contribution of the network elements to the diffusion capability of the peat samples was generally well reproduced. We can therefore state that the effective performance of the simulation method was rather good.

In our opinion, a thorough sensitivity analysis of the geometrical structure of the well-founded and established methodology of pore network modeling is beyond the scope of this study. The study contains the generation and assessment of an experimental dataset of peat soil gas diffusivity and the assessment of the performance of the pore network modeling methodology considering the reproduction of the measurement results. We have revised the discussion by further assessing the applicability of the simplified pore geometry used in the gas diffusivity simulations with respect to the actual structure of peat.

L408–423: "PNM has been extensively and successfully applied to the simulation of fluid flow and mass transport in porous media (e.g., Blunt et al., 2002, de Vries et al., 2017, Sadeghi et al., 2020). If the features and phenomena relevant to the simulated process are adequately identified and described, PNM is able to give reliable results (Xiong et al., 2016). Several features important in gas diffusion were well accounted for in our simulations. The topology and connectivity of the pore network, which was obtained directly from the µCT images, corresponded to that of the macropore space of the peat sample. The structural anisotropy of the pore space, which is a conspicuous characteristic of peat (McCarter et al., 2020), was incorporated into the topology of the pore network, as the orientation of each transport conduit was represented by the orientation of the throat in the three-dimensional pore network. As in Eq. (8), the diffusion rate between two pores is dependent on the length and the cross-sectional area of the conduit between them. The segmentation of the pore space to individual pores was not based on unequivocally determined criteria, which is always the case for soil (Nimmo, 2005), but the spatial coordinates and the cross-sectional areas of the resulting pores and throats were well-defined. The watershed algorithm used in pore segmentation is also well suitable for high-porosity materials, such as peat (Gostick, 2017). The simplification of the shapes of the transport conduits into spheres and cylinders was therefore the only major approximation that may have considerably affected the calculated diffusional conductance values of the conduits. In addition, the minimum conduit diameter of 100 µm was 3 orders of magnitude higher than the mean free path of air molecules at standard conditions (Rumble, 2021), and the Fickian diffusion approach remained valid."

**Results part:**

line 251 following: The database is weak and results do not contain surprising or interesting patterns, so strongly shorten!

There are few studies on gas diffusivity in peat (other studies are discussed in Sect. 4.1), and the sample sizes were generally very small in these studies. In comparison to these previous studies, our sample size is not very small. We consider that the experimental results presented for gas diffusivity in peat provide a relevant scientific contribution, and the assessment of the results is thus justified.

The variability of peat properties due to its heterogeneous physical structure is also highlighted in the analysis of the results. However, we have slightly shortened the paragraph.

Shrinkage is a very critical issue but is completely disregarded in the measurements. I would suggest to correct epsilon for the reduced total volume.

The results for shrinkage are presented in Kiuru et al. (2022). The decrease in sample height was taken into account in the calculation of air-filled porosity and the soil gas diffusion coefficient at -10 kPa conditions. The description in Sect. 2.2 is therefore imprecise. We have added information on the measured sample shrinkage and on the observations of shrinkage properties in the Results section. We have also corrected the text in Sect. 2.2 by mentioning the facts that the air-filled porosity was calculated from the reduced total volume and the soil gas diffusion coefficient was calculated using the reduced sample height at -10 kPa conditions.

L268–271: "The shrinkage of the samples in the vertical direction at –10 kPa matric potential was on average 6.3 % in the top layer, 3.7 % in the middle layer, and 2.3 % in the bottom layer. The vertical shrinkage was considered to be negligible at higher-matric-potential conditions. In addition, slight horizontal shrinkage was observed in some of the middle-layer and bottom-layer samples at –10 kPa conditions. The estimated decrease in the sample diameter was generally of the order of 1 to 2 % or less."

L113–115: "The bulk density of a peat sample was determined from its dry mass and the saturated volume, and the volumetric water contents at different matric potentials were also calculated in relation to the saturated volume with the exception that the reduction of the sample volume due to shrinkage was taken into account at -10 kPa conditions."

L138–139: "The decrease in the sample length due to shrinkage was taken into account in the calculation at -10 kPa conditions."

Table 1: two digits are sufficient

We would prefer to present the air-filled porosity values with three significant figures as we have done for the volumetric water content and bulk density of the same samples in Kiuru et al. (2022) and for the soil gas diffusion coefficient in this manuscript.

Table 2: "significant differences" between what?

We meant that different letters in the table indicate a significant difference in the soil gas diffusion coefficient between different peat sampling depths. The same sentence structure is used in the footnote of Table 1, and we have clarified both occasions.

Table 1 footnote: "Different letters indicate a significant difference in air-filled porosity between different sampling depths (p

Blunt, M. J., Bijeljic, B., Dong, H., Gharbi, O., Iglauer, S., Mostaghimi, P., Paluszny, A., and Pentland, C.: Pore-scale imaging and modelling, Adv. Water Resour., 51, 197–216, https://doi.org/10.1016/j.advwatres.2012.03.003, 2013.

Kiuru, P., Palviainen, M., Grönholm, T., Raivonen, M., Kohl, L., Gauci, V., Urzainki, I., and Laurén, A.: Peat macropore networks – new insights into episodic and hotspot methane emission, Biogeosciences, 19, 1959–1977, https://doi.org/10.5194/bg-19-1959-2022, 2022.